# How Many Parameters are Needed to Represent Polar Sea Ice Surface Patterns and Heterogeneity?

Joseph Fogarty[1], Elie Bou-Zeid[1], Mitchell Bushuk[2], and Linette Boisvert[3]

[1]Civil and Environmental Engineering Department, Princeton University, Princeton NJ, USA
[2]National Oceanic and Atmospheric Administration, Geophysical Fluid Dynamics Laboratory, Princeton NJ, USA
[3]Cryospheric Sciences Lab, NASA Goddard Space Flight Center, Greenbelt MA, USA

**Correspondence:** Elie Bou-Zeid (ebouzeid@princeton.edu)

**Abstract.** Sea ice surface patterns encode more information than can be represented solely by the ice fraction. The aim of this paper is thus to establish the importance of using a broader set of surface characterization metrics, and to identify a minimal set of such metrics that may be useful for representing sea-ice in Earth System Models. Large-eddy simulations of the atmospheric boundary layer over various idealized sea ice patterns, with equivalent ice fraction and average floe area, demonstrate that the spatial organization of ice and water can play a crucial role in determining boundary-layer structure. Thus, different methods to quantify heterogeneity in categorical lattice spatial data, such as those used in landscape ecology and Geographic Information System (GIS) studies, are used here on a set of high-resolution, recently-declassified sea ice surface images. It is found that, in conjunction with ice fraction, the patch density (representing the fragmentation of the surface), the splitting index (representing the variability in patch size), and perimeter-area fractal dimension (representing the tortuosity of the interface) are all required to describe the two-dimensional pattern exhibited by a sea ice surface. For surfaces with anisotropic patterns, the orientation of the surface relative to the mean wind is also needed. Finally, scaling laws are derived for these relevant landscape metrics to estimate them from aggregated spatial sea ice surface data at any resolution. The methods used and the results gained from this study are a first step towards further development of methods to quantify the variability of polar sea surfaces, and for parameterizing mixed ice-water surfaces in coarse geophysical models.

## 1 Introduction

The polar sea ice surface, a sensitive indicator of global climate change, shows persistent biases in sea ice fraction and extent in coarse-resolution Earth System Model (ESMs) (Liu et al., 2022; Casagrande et al., 2023; Myksvoll et al., 2023). Among other causes, these biases result from the inability of ESMs to resolve the fine-scale spatial variability of sea ice, and the resulting exchanges with the ocean below (Ramudu et al., 2018) and atmosphere aloft (Bates et al., 2006; Esau, 2007). The effect of this subgrid scale sea ice variability is typically parameterized in climate models using the ice fraction, $f_i$, to determine surface-atmosphere fluxes for an equivalent surface that would produce the same grid-cell averaged exchanges as the ice-water mixture. Usually, either an equivalent homogeneous surface or mosaic flux aggregation are used (Elvidge et al., 2016; Bou-Zeid et al., 2020; Elvidge et al., 2021), but both yield an average flux weighted by the ice and water fractions that is inaccurate as it does not account for the impact of surface heterogeneity on the dynamics of the lower atmosphere and the nonlinear interactions

with the air flow above the ice and water (de Vrese et al., 2016; Lüpkes et al., 2012). This incomplete representation of sea ice surface and boundary-layer structure then results in errors in the turbulent exchanges of heat, moisture, and momentum across polar sea ice surface (Nilsson et al., 2001; Bourassa et al., 2013; Taylor et al., 2018). The dynamics and secondary circulations below the first vertical grid cell level in climate models are particularly under-resolved, and they have a direct impact on air-surfaces exchanges; it is thus imperative to understand how these features influence fluxes (Mahrt, 2000; Essery et al., 2003; de Vrese et al., 2016). These gaps in representing fine scale dynamics and fluxes propagate to the projection of future changes in the Arctic climate system and resulting surface energy budget (Persson et al., 2002; Miller et al., 2017), which may be one reason why climate model ensembles consistently underpredict Arctic sea ice sensitivity to surface temperature warming. This underprediction has persisted throughout the last three Intergovernmental Panel on Climate Change (IPCC) model development cycles (Stroeve et al., 2007; Rosenblum and Eisenman, 2016, 2017; Notz and Community, 2020). The resulting uncertainty in climate models' ability to predict sea ice future evolution hinders effective action and decision-making; therefore, improving these models is imperative (Notz and Stroeve, 2018; Docquier and Koenigk, 2021).

The fringe zone that separates densely consolidated sea ice from the open ocean is known as the marginal ice zone (MIZ) (see Dumont (2022) for a review on the current state of MIZ research). In the MIZ, the sizes and organization of sea ice floes and water are influenced by winds, sea currents, waves, and material ice properties (Wang et al., 2016; Ren et al., 2021; Herman et al., 2021; Hwang and Wang, 2022). What makes this region unique is that the near-surface air temperatures may fall in between the surface temperatures of the sea ice and water, resulting in abrupt spatial transitions between stabilizing and destabilizing surface buoyancy fluxes (Lüpkes et al., 2012). Such transitions produce drastically different turbulence-mean non-equilibrium dynamics and time scales (as shown for comparable land-water transitions by Allouche et al. (2021, 2023b)), all affecting the surface-atmosphere exchanges between the air, water, and sea ice. The ice fraction $f_i$ in the MIZ is between 15% and 80% (Strong et al., 2017); however, any region of fractured sea ice gives rise to these abrupt transitions. It is precisely in these regions where the linear weighted averaged approaches described above will be most inadequate, and where the surface transitions will play a key role in the dynamics. Thus, it is important to devise better methods to quantify the heterogeneity of a surface, characterize its patterns, and encode this information in coarse-resolution ESMs to better represent the polar environment.

To that end, the complex geometric patterns formed by sea ice floes need to be analyzed. Larger floes will have proportionally more of an effect on the surface-atmosphere fluxes, but smaller floes, with more frequent transitions, will exacerbate the non-linearity of the exchange processes. These surface-atmosphere fluxes impart a large effect on the atmospheric boundary layer (ABL) overlaying the marginal ice zone (MIZ-ABL). As a thought experiment, consider an ice-water surface with a very fine checkerboard pattern, and a sea ice fraction of $f_i = 0.5$; this configuration will lead to statistically-homogeneous ice floes that are locally variable at the surface, but are effectively homogeneous in regards to the MIZ-ABL where turbulence will rapidly mix their small-scale signatures (Brutsaert, 2005; Mahrt, 2000; Bou-Zeid et al., 2004). However, two large patches of sea ice and water (meso-$\alpha$ heterogeneity, see Bou-Zeid et al. (2020)), also with a sea ice fraction of $f_i = 0.5$, will develop a large circulation closer to that of a sea breeze due to the abrupt transition between two large homogeneous surfaces (Porson et al., 2007; Crosman and Horel, 2010; Allouche et al., 2023a). The dynamics and thermodynamics in this MIZ-ABL system, and

the surface exchange therein, will thus be quite different over these two patterns even if some key surface properties, e.g. temperature and roughness, are identical (Bou-Zeid et al., 2007).

Given its importance and the challenges outlined above, previous work has attempted to quantify the heterogeneity of sea ice surfaces (Wenta and Herman, 2018, 2019; Michaelis et al., 2020; Horvat, 2021; Dumont, 2022) utilizing surface and meteorological properties such as sea ice fraction, geostrophic velocity, lead width, or floe size distribution. Furthermore, parameterizations for flow over leads in sea ice have been developed based on non-eddy resolving models (Lüpkes et al., 2008; Michaelis et al., 2021). Michaelis and Lüpkes (2022) have also conducted turbulence parameterizations (based on large-eddy simulation models) over ensembles of leads, but with a two-dimensional ice fraction geometry and a higher ice fraction. However, the small-scale patterns in the MIZ, especially as the resolution is increased, require broader and more versatile methods of heterogeneity characterization (e.g., Mandelbrot (1967)). In addition, the computational grid of even the highest-resolution weather or climate models models cannot resolve all the spatial features in the MIZ. One thus needs to consider how to represent, in such models, unresolved surface characteristics that can be thought of as lattice-type spatial structures, defined by Cressie (1993). Observational data of MIZ ice patterns also have a finite resolution and are thus comparable to lattice data, which then allows one to utilize different metrics specifically defined for lattice surfaces that offer ways to characterize the heterogeneity patterns of these Polar surfaces. In this paper, we examine approaches for this quantification commonly used in landscape ecology, a field that has generated a multitude of ways to study lattice spatial data (Li and Reynolds, 1994, 1995; Pickett and Cadenasso, 1995).

Studies in landscape ecology have previously searched for an optimal independent group of metrics to be used in understanding the heterogeneity of lattice surfaces. Riitters et al. (1995) used a multivariate factor analysis to suggest six groups of metrics, including image texture, average patch compaction, and average patch shape. Cushman et al. (2008) used principal component analysis to suggest seven broad metrics at the landscape level, including contagion, large patch dominance, and proximity (see Table 9 in that study). For the two-dimensional binary sea ice-water surfaces considered in this study, we chose the variance inflation factor (VIF) technique to reduce these metrics to a compact set that are weakly dependent on one another to minimize information redundancy (Miles, 2014).

The questions that will be answered in this study are:

1. Is the sea ice fraction of a MIZ surface, combined with some measure of average floe area, sufficient to predict the behavior of the overlying MIZ-ABL?

2. If not, what other surface information in a two-dimensional lattice spatial pattern is needed to describe air-sea interaction?

3. How can this surface information be applied to sea ice surfaces in weather models and ESMs, considering factors such as availability of information, resolution-resampling invariance, and ease of understanding?

Section 2 will detail the methods on the idealized and real-world maps used in this study; this includes the steps taken to reduce multicolinearity and determine which landscape metrics, alongside sea ice fraction, give additional information on the pattern of the sea-ice surface. The large-eddy simulations that will be used are also presented in this section, but a more

complete description is found in Appendices A and B. Section 3 will report the results of the idealized sea ice surfaces in the large-eddy simulation, thus answering Question 1, and motivating the remaining questions. Section 4 will report the results from the 2D surface analysis, with additional discussion in Section 5 on principal directions and climate model implications, thus answering Questions 2 and 3. Section 6 will synthesize the findings and outline open questions that can guide future investigations of sea ice heterogeneity.

## 2 Methods and Data

### 2.1 Large-Eddy Simulations

Large-eddy simulations (LES) of the MIZ-ABL over different idealized configurations of sea ice were conducted. LES are widely used to model heterogeneous high Reynolds number flows (Baidya Roy, 2002; Bou-Zeid et al., 2004) in convective boundary layers (Courault et al., 2007; Maronga and Raasch, 2013), stable boundary layers (Huang et al., 2011), and coastlines (Allouche et al., 2023a) to name a few; see Section 3.6 of Stoll et al. (2020). This heterogeneous high-Reynolds number description aptly applies to the MIZ-ABL. Unlike a direct numerical simulation (DNS), an LES is able to attain Reynolds numbers representative of the MIZ-ABL (Re $\sim 10^7$), because the smaller turbulent eddies (smaller than the filter size, which is comparable to the numerical grid spacing in our simulations) are not explicitly resolved. However, unlike Reynolds averaged Navier-Stokes (RANS) approaches, which encompass all weather and climate models, LES directly resolves and captures the large turbulent eddies, the heterogeneity of the surface, advective fluxes, and the large-scale sea ice patterns, making it a computationally and physically appealing approach for the problem at hand. By retaining these larger structures, most of the turbulent energy and fluxes are explicitly resolved, allowing for investigation of three-dimensional flow structures that may arise over these heterogeneous surfaces.

LES is thus used to model MIZ-ABL flow over $10\,\mathrm{km} \times 10\,\mathrm{km}$ patterns of idealized ice/water surfaces, modulated by a Coriolis force at the latitude of $\Phi = 90°$ N (therefore a Rossby number of Ro = 13.7) in a horizontally periodic domain (see figure1) . We note that simulations at the same $Ro$, even if at lower latitudes and mean wind speeds, would give similar results (see dimensional analysis of flow over heterogeneous surfaces in Fogarty and Bou-Zeid (2023) and Allouche et al. (2023a)). This full domain is smaller than a single grid cell in state-of-the-art ESMs, underlining why the simulated properties of the MIZ-ABL in ESMs require sub-grid scale (SGS) parameterization. For leads of $\sim 1\,\mathrm{km}$ width, a grid spacing of $10-20\,\mathrm{m}$ is usually chosen after grid convergence tests (Lüpkes et al., 2008; Gryschka et al., 2023). Our choice of a coarser horizontal resolution of $100\,\mathrm{m}$ reflects our aim to "zoom out" from a typical lead and examine the MIZ as a whole (which is also reflected in the horizontally periodic nature of the domain). A finer resolution in our simulations, while possibly improving the representation of turbulence and plume dynamics, may sacrifice some of the large scales and secondary circulations that arise from these heterogeneous surfaces that we aim to capture with a large domain. The coarse resolution also will not compromise our ability to answer the first question because, as we show later, the differences in the dynamics between simulations with identical ice fractions but different surface patterns are significant and far exceed any plausible impact of grid resolution. The simulations in this section are meant to demonstrate the need for surface analysis, given a constant ice fraction and average ice floe area,

**Table 1.** Large-eddy simulation numerical details. Time is represented in terms of inertial periods, $2\pi/f_c$, which is the time scale associated with the response of the mean flow since it represents the Coriolis redistribution of energy between $u$ and $v$ (Momen and Bou-Zeid, 2016)

| | |
|---|---|
| Domain height, $z_i$ | $1\,\mathrm{km}$ |
| Horizontal domain size, $L_x \times L_y$ | $10\,\mathrm{km} \times 10\,\mathrm{km}$ |
| Number of grid points $(N_x, N_y, N_z)$ | $(100, 100, 50) \approx 5 \times 10^5$ points |
| Vertical mesh spacing, $dz$ | $20\,\mathrm{m}$ |
| Horizontal mesh spacing, $dx$, $dy$ | $100\,\mathrm{m}$ |
| Initial air potential temperature $\theta_{a,0}$ | See Appendix B, constant profile |
| Coriolis parameter $f_c$ | $1.46 \times 10^{-4}\,\mathrm{s}^{-1}$ |
| Warm-up period | 5 inertial periods ($10\pi/f_c$) |
| Averaging period | 1 inertial period ($2\pi/f_c$) |
| Simulation time step | $0.05\,\mathrm{s}$ |
| Frequency of statistical sampling | 100 timesteps = $5\,\mathrm{s}$ |

and we thus do not focus on the quantitative aspect of the output. See all details for the simulations in this study in Table 1; more details on the numerical aspects of our LES are described in Appendix A.

The bottom boundary condition for each simulation can be thought of as categorical lattice spatial data, where each node represents either the "ice" or "water" class. An "ice" node is prescribed a surface temperature of $\theta_i = 255\,\mathrm{K}$, typical of autumn and spring temperatures in the Central Arctic, and a momentum roughness length of $1\,\mathrm{mm}$, while a "water" node is prescribed a surface temperature of $\theta_w = 271\,\mathrm{K}$, roughly the freezing point of seawater, and a roughness length of $1\,\mathrm{cm}$. The heat roughness length is set to $0.1\,\mathrm{mm}$ for the entire surface, ice and water. There is a very large variability in roughness lengths in Arctic sea ice simulations (see Lüpkes et al. (2008); Andreas et al. (2010); Lüpkes et al. (2012); Elvidge et al. (2016); Gryschka et al. (2023)) that reflects the physical differences between a calm and a stormy sea and between new flat ice and old deformed and refrozen ice. However, our choice of roughness lengths prioritized keeping constant ratios (which are the key dimensionless parameters that influence the results Fogarty and Bou-Zeid (2023); Allouche et al. (2023a)) among simulations to focus on the effect of sea ice patterns. Results with different values but the same ratios will lead to almost identical conclusion. In addition, sensitivity analyses not shown here indicated that the roughness lengths, even when their ratios are changed, had a very minor impact on the quantitative results compared to the temperature contrast, ice fraction, and ice-water patterns . The initial air potential temperature $\theta_{a,0}$, a constant profile, is defined such that the area-averaged sensible heat flux as computed by the Monin-Obukhov surface flux parametrizations is zero, and thus lies between that of the ice and water surface temperatures (see Appendix B for details). The LES-modeled heat flux will, however, not be zero.

All five patterns, displayed in figure 2, were simulated. They have a fixed sea ice fraction of $f_i = 0.46$ and mean floe area of $11.56 \times 10^6\,\mathrm{m}^2$. The geostrophic wind ($M_g = 2\,\mathrm{m\,s}^{-1}$) flows left-to-right at an angle of $18°$ relative to the $x$-axis in all simulations, expected to give a surface wind $M_0$ that is roughly aligned with the $x$-axis for homogeneous neutral surfaces due to Ekman veer (Ghannam and Bou-Zeid, 2021). These simulations are dry runs with no clouds present. The turbulence field

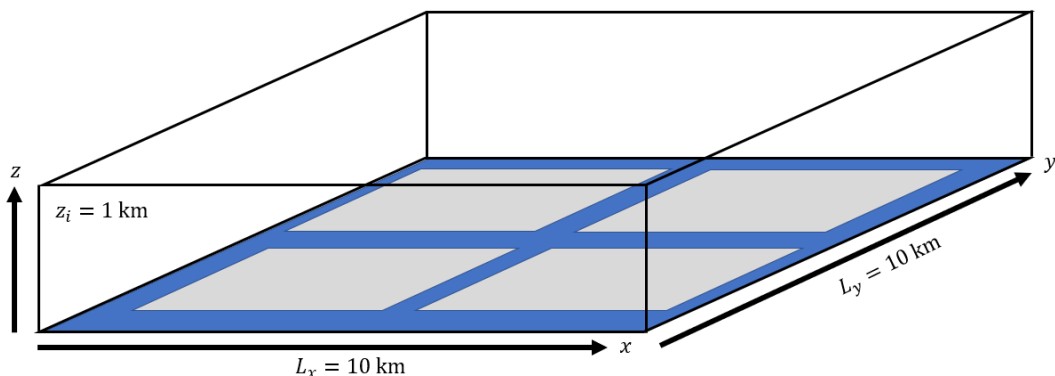

**Figure 1.** Schematic of the large-eddy simulation domain set-up. The ice (in grey) has a surface temperature of $\theta_{0,i} = 255\,\mathrm{K}$ and roughness length $z_{0,i} = 1\,\mathrm{mm}$; the water surface (in blue) has $\theta_{0,w} = 271\,\mathrm{K}$ and $z_{0,w} = 1\,\mathrm{cm}$. The bottom boundary represents one of the many cases (Pattern1) illustrated in Figure 2.

is warmed up for about 60 hours, and the statistics are then Reynolds-averaged over an additional 12 hours. A variable with an overbar denotes averaging in time, used as a surrogate for ensemble Reynolds averaging, and any spatial averaging over the heterogeneous domain in $x$ and $y$ will be denoted by angled brackets.

In addition to the Rossby number and the roughness ratio discussed earlier, an important dimensionless input parameters in these simulations is the heterogeneity Richardson number defined as

$$\mathrm{Ri}_h = \frac{g}{\theta_{a,0}} \frac{\theta_w - \theta_i}{M_g^2 / z_i}. \tag{1}$$

This $Ri$ encodes the competition between buoyancy driven circulations generated by the surface temperature contrast and the uniform flow that would result from the synoptic forcing $M_g$. All simulations in this study have equivalent inputs of $Ri_h$, $Ro$, 155    and $z_{0,w}/z_{0,i}$. Based on previous dimensional analyses and LES of flow over heterogeneous surfaces (Omidvar et al., 2020; Allouche et al., 2023a; Fogarty and Bou-Zeid, 2023), matching these dimensionless inputs is required to be able to focus on the effect of surface sea ice fraction and patters.

## 2.2 Ice Map Data

While the LES utilizes idealized surfaces to examine the influence of patterns on the MIZ-ABL, examining what other land-160    scape metrics might be important for surface characterization necessitates using real-world sea-ice maps. The lattice spatial data that will be used in the statistical analysis (see Section 2.3) are derived from recently declassified high-resolution (1 m) national technical means (NTM) literal image derived products (LIDPs), detailed in Kwok (2014). These images we use here already underwent a supervised maximum likelihood classification algorithm, which assigned either a water or ice surface class (Fetterer and Untersteiner, 1998; Fetterer et al., 2008) for each pixel in the original LIDP. This process converted the

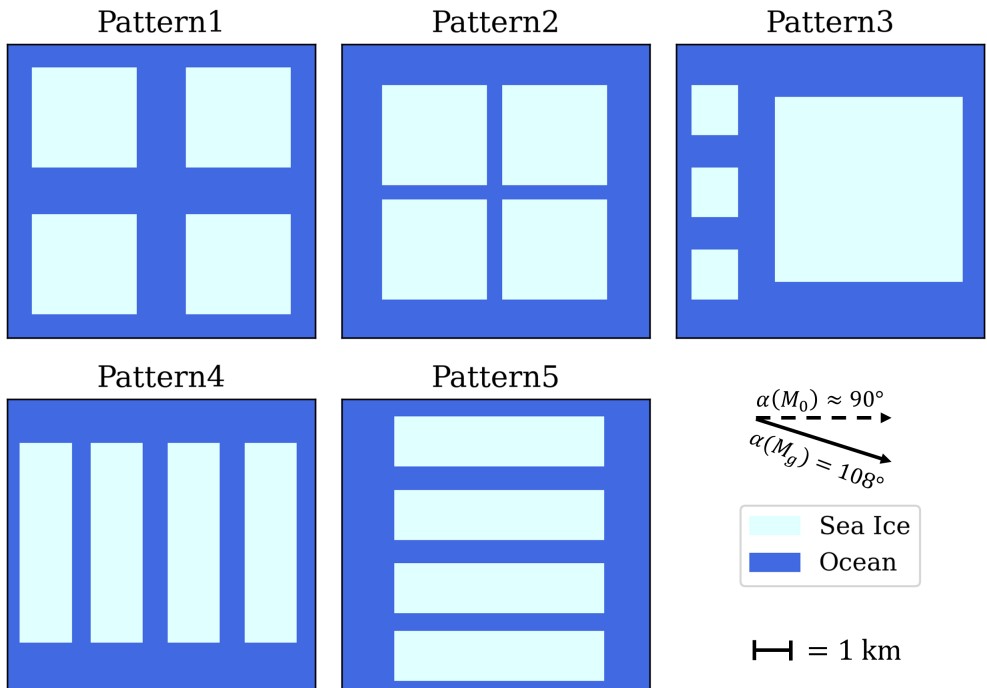

**Figure 2.** Birds-eye view of the five idealized $10\,\text{km} \times 10\,\text{km}$ sea ice surfaces surfaces created for the large-eddy simulation. The geostrophic wind ($M_g$) flows at an angle of $108°$ such that the near-surface winds ($M_0$) flows from left-to-right in all patterns. The results from the LES over these five patterns are discussed in Section 3.

high-resolution LIDPs into categorical lattice spatial data (where each cell represents one of two possible surface types, ice or water).

These maps, which have a horizontal extent of up to $10\,\text{km}$ by $10\,\text{km}$, comprise the dataset used to calculate landscape metrics. Some of the images did not fully cover this full extent, and thus in order to retain the real-world sea ice geometry, we "reflected" this onto the areas of no data. All metric calculations and analyses have been done on these modified surfaces. The advantage of this high-resolution large-extent data set is that we can analyze how these metrics change with grain size. These maps are thus aggregated from $1\,\text{m}$ resolution to $2\,\text{m}$, $10\,\text{m}$, $20\,\text{m}$, $50\,\text{m}$, $100\,\text{m}$, $200\,\text{m}$, $500\,\text{m}$, $1\,\text{km}$, and $2\,\text{km}$ resolutions; resampling was done using the nearest-neighbor method in the Python Imaging Library (PIL). These resolutions cover common resolutions used from fine large-eddy simulations to numerical weather prediction (NWP) models. Due to excessive computational processing time for the original $1\,\text{m}$ resolution data, the highest resolution at which landscape metrics were calculated was $2\,\text{m}$.

One important caveat of this dataset is that the histogram of sea ice fraction $f_i$ is heavily skewed towards higher sea ice fractions (see figure 3). We recognize that this may lead to bias in the results; however, the analysis methods developed here are insensitive to $f_i$ and can certainly be applied to other datasets with more uniform sea ice fraction distributions in the

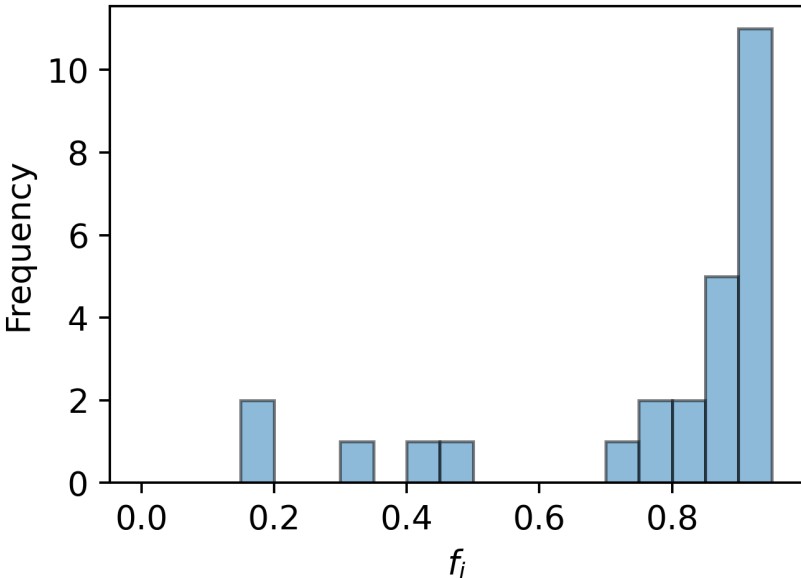

**Figure 3.** The distribution of $f_i$ values in the Fetterer et al. (2008) ice map dataset.

future. The high resolution afforded by the present dataset remains a key appealing factor in adopting it for the present study
since it allows aggregating and tracking how landscape metrics change from actual surface states when resampled to a coarser
numerical grid.

### 2.3 Landscape Metric Space Reduction

The FRAGSTATS spatial analysis program was used to calculate the landscape metrics (McGarigal and Marks, 1995) based
on the lattice spatial data. Given a GeoTIFF-format raster lattice data, FRAGSTATS will calculate the patch metrics, class
metrics, and landscape metrics of your choice. Patch metrics are computed individually for every patch in the landscape, and
are thus not relevant for the current study of sea-ice surface patterns. Class metrics are computed for every patch type (class)
in the landscape. In this study, that would mean calculating metrics for sea ice only and water only, which may be useful in
other applications of pattern analysis; however, for this study, we want to look at the aggregate patterns of sea ice and water
combined. Thus, only landscape metrics were calculated. Sea ice fraction (calculated as the number of cells of ice type divided
by the total number of ice and water cells) was calculated using PIL, the only metric not calculated by FRAGSTATS.

This resulted in 22 landscape metrics that focus on the global patterns of the surface. Many of these landscape metrics,
however, are correlated with one another; for example, patch density and mean patch size are proportional to one another. This
is due to the fact that there are limited observations one can make about a surface (number of patches, area of a patch, amount

of edge in a patch, etc.), yet an infinite number of operations one can perform on them. Many of these metrics (especially at the landscape level) are thus simply different ways to aggregate or statistically analyze these observations.

While colinearity between two metrics can be easily detected through a correlation matrix, multicolinearity (when one indicator is a linear combination of two or more other indicators) is more likely in these types of data sets. It is thus possible for two or more landscape metrics to jointly define another metric. An objective and statistical way of reducing these parameters is hence needed. Here, we chose the variance inflation factor (VIF),

$$VIF_i = \frac{1}{1 - R_i^2},$$
(2)

where $R^2$ is the coefficient of determination, to detect multicolinearity of these heterogeneity parameters (see Ibidoja et al. (2023)). For each metric $X_i$, where $i \in [1, ..., 22]$, the VIF was calculated over the regression equation

$$X_i = \alpha_0 + \alpha_{i+1}X_{i+1} + \alpha_{i+2}X_{i+2} + ... + \alpha_{21}X_{21}$$
(3)

where $\alpha_0$ is a constant. The `statsmodels` Python library (Seabold and Perktold, 2010) was used for these computations. The metric with the largest VIF was then removed from the dataset, and thus not considered to be important in the quantification of sea ice surfaces. All VIFs were then recalculated for this 'reduced' dataset, and the new metric with the highest VIF was removed. Through this process, metrics are removed one by one until all remaining metrics exhibit a VIF less than a pre-defined cutoff, which was set as $VIF < 2.5$. While this low of a cutoff may not be necessary in certain practices of multicolinearity reduction (O'Brien, 2007), the ultimate goal of this technique is to reduce the parameter space. In other words, for climate modelers, a lower amount of metrics in their SGS parameterizations result in a more practical models.

Each of these metrics, listed in Table C1, can be clustered into one of six "metric groups": *Area and Edge*, *Shape*, *Core Area*, *Aggregation*, *Contrast*, and *Diversity*. The first four metrics are important in a sea ice surface. *Area and Edge* metrics deal with the size of floes and the amount of edge they create, while *Shape* metrics discriminate based on patch morphologies and overall geometric complexity. *Core Area* metrics analyze the area within a patch beyond some specified buffer width. An *Aggregation* metric will focus on the tendency of patches of similar types to be spatially aggregated in the landscape, or otherwise dispersed.

The last two metric groups, *Contrast* and *Diversity*, are less important for the present application to sea ice. *Contrast* metrics refers to the magnitude of difference between adjacent patch types with respect to some attribute - in the case of a sea ice surface, with only two classes (ice and ocean), there is only one 'contrast' between two categories, and thus metrics in this group are simply represented by the contrast of surface temperature and roughness. *Diversity* metrics are influenced by the number of patch types present and the area-weighted distribution of those patch types. In this case we only have two types of patches (ice and water) so the diversity is the same in all maps, and their weighted distribution is related to the ice fraction. Further information on all of these metric groups can be found in the FRAGSTATS manual (McGarigal and Marks, 1995).

## 3 Results: the MIZ-ABL over Idealized Configurations

The large-eddy simulation technique detailed in Section 2.1 was used to simulate the MIZ-ABL over different configurations of sea ice patterns. Figure 4 displays the Reynolds- and horizontally-averaged normalized vertical profiles of the horizontal

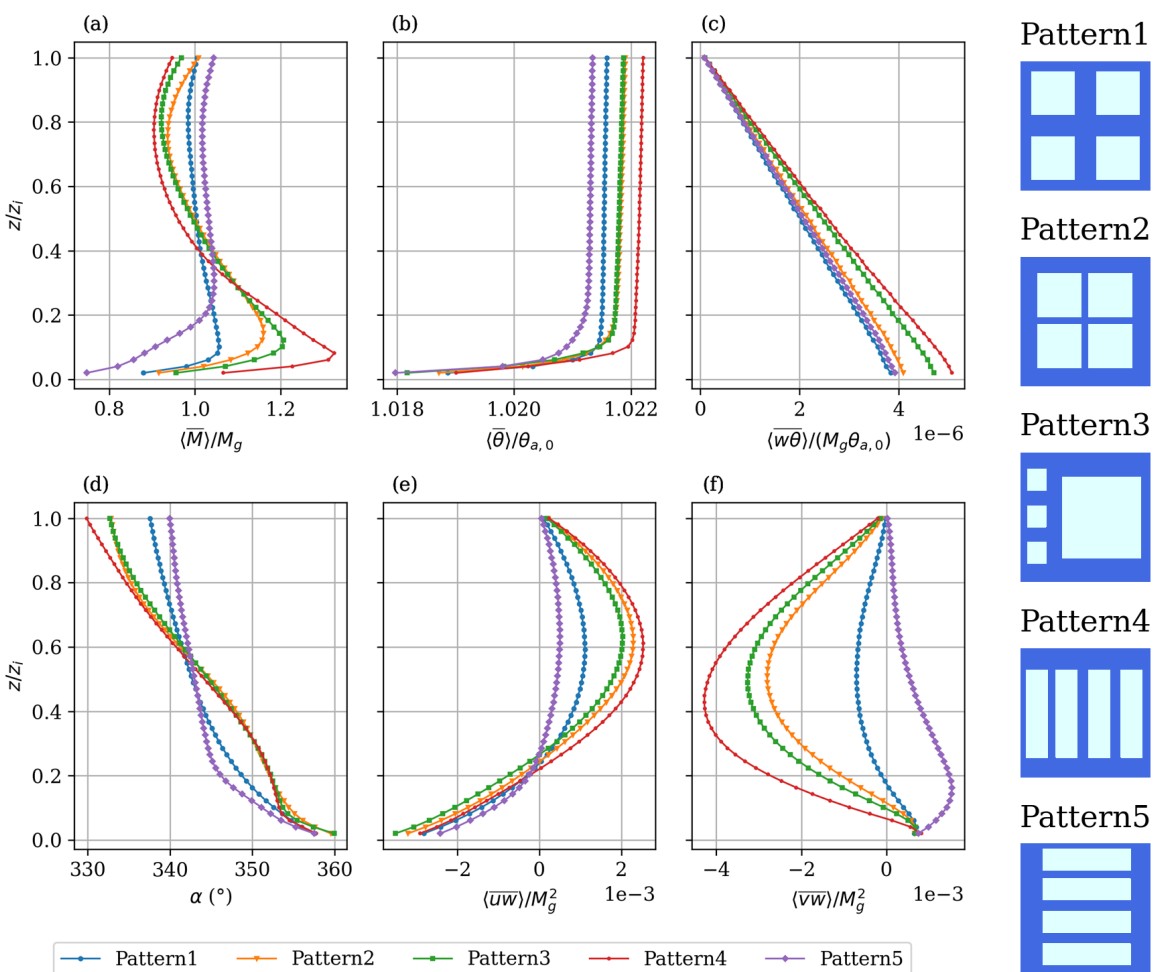

**Figure 4.** Vertical profiles of normalized (a) horizontal wind speed, (b) potential air temperature, (c) total heat flux, (d) geostrophic wind direction $\alpha$, (e) total stress in the streamwise direction, and (f) total stress in the cross-stream direction, for all five patterns.

wind speed $M = \langle \sqrt{u^2 + v^2} \rangle$ (normalized by geostrophic velocity $M_g$), potential air temperature (normalized by the initial potential air temperature $\theta_{a,0}$), total heat flux $\langle \overline{w\theta} \rangle$ (normalized by $M_g\theta_{a,0}$), and total horizontal stresses $\langle \overline{uw} \rangle$ and $\langle \overline{vw} \rangle$ (both normalized by $M_g^2$). Note that these lump the turbulent fluxes with the dispersive fluxes that arise over heterogeneous surfaces from the spatial correlation of the mean (time-averaged) fields (Raupach and Shaw, 1982; Finnigan and Shaw, 2008; Li and Bou-Zeid, 2019). The results clearly display significant differences, underlining the fact that sea ice fraction is not a sufficient surface metric to describe MIZ-ABL dynamics, even in these simulations where the mean floe size was also kept constant.

In all five simulations, the largest difference occurs between Pattern4 and Pattern5 (the red and purple lines, respectively), which is to be expected since the geostrophic wind, and thus the near-surface wind, is flowing parallel vs. perpendicular to the

strips of ice (Willingham et al., 2014; Anderson et al., 2015; Salesky et al., 2022; Fogarty and Bou-Zeid, 2023). Although wind
direction is not a surface property of the sea ice, its orientation relative to that of the surface features is still an important driver
that must be taken into consideration and will be discussed further on. All patterns developed a low-level jet (LLJ), which can
be seen in Figure 4a, though the LLJs in Pattern1 and Pattern5 are weak (Tetzlaff et al., 2015; Michaelis et al., 2021). The LLJs
seem to increase in Pattern2 and Pattern3, likely due to large swaths of ice in the direction of the geostrophic wind; the unstable
to stable transitions in these ice regions decouple the air from the surface friction over the long stable ice patches, allowing
low-level acceleration of the wind. This mechanism is similar to the one advanced by Blackadar (1957) for the creation of a
low level jet via an inertial oscillation in time as the ABL transitions to a stable regime at sunset. However, in this case the
oscillation is in space as columns of air advect from a hot to a cold surface and decouple from the surface. The second strongest
LLJ is in Pattern3, and is likely due to the one large ice floe, and would persist for any wind direction (the same can be said
about Pattern2). The strongest LLJ in Pattern4 is also likely reinforced by the secondary circulations (not shown) that consist
of streamwise aligned rolls driven by the lateral contrast in surface temperature, and by the fact that the small strips of ocean
between the ice floes are not wide enough to 'interrupt' the circulations. However, unlike Pattern2 and Pattern3, Pattern4 is
highly anisotropic, so the geostrophic wind direction is of more importance in this case, as can be deduced from the fact that
Pattern5 has no LLJ despite being simply a 90° rotated version of Pattern4.

Each simulation saw an increase in potential temperature by 2.1-2.2%; this warming is consistent with the large ocean
fraction in each of the patterns. Despite initializing the air temperature to produce zero fluxes following the Monin-Obukhov
flux models that assume equilibrium of the air and water above each surface grid cell, the upward fluxes over the warm water
are larger than the downward one over cooler ice due to the effect of advection that perturbs the equilibrium.

Major differences are also seen in the total streamwise and cross-stream stresses, displayed in Figure 4d and e, respectively.
Again, Pattern4 and Pattern5 exhibit the highest differences from one another due to the geostrophic wind direction. All
simulations have a similar negative streamwise stress in the surface layer, but at higher altitudes in MIZ-ABL, the differences
between simulations are greater. Above the LLJ, some of the stresses turn positive implying upward transfer of momentum
from the LLJ, explaining differences that may be seen below the blending height (Wood and Mason, 1991; Mahrt, 2000;
Brunsell et al., 2011). The cross-stream components of the total momentum flux, seen in 4e, are all quite distinct from one
another indicating significant differences in the wind and stress Ekman rotation with height.

To further explain the differences seen in these patterns, we consider the decomposition of the total flux to its dispersive
and turbulent contributions. Vertical turbulent heat flux is designated as $\overline{w'\theta'}$, while vertical turbulent streamwise and cross-
stream stress are designated as $\overline{u'w'}$, and $\overline{v'w'}$, respectively. Dispersive fluxes emerge in a time-averaged but spatially-variable
mean flow (Raupach and Shaw, 1982; Li and Bou-Zeid, 2019). Since our Reynolds averaging is done in time, we can spatially
decompose any Reynolds-averaged variable following (for the vertical velocity as an example) $\overline{w} = \langle \overline{w} \rangle + \overline{w}''$, where the
brackets represent the spatial average (as defined in Section 2.1) and the double-prime represents the variations of the mean

planar fields in space. We then calculate the local dispersive fluxes, using vertical heat flux as an example, by:

$$\overline{w}\overline{\theta} = \left(\langle\overline{w}\rangle + \overline{w}''\right)\left(\langle\overline{\theta}\rangle + \overline{\theta}''\right) \tag{4}$$

$$= \langle\overline{w}\rangle\langle\overline{\theta}\rangle + \overline{w}''\langle\overline{\theta}\rangle + \langle\overline{w}\rangle\overline{\theta}'' + \overline{w}''\overline{\theta}''. \tag{5}$$

We then spatially average the entirety of Equation 5 over the horizontal plane to obtain

$$\langle\overline{w}\overline{\theta}\rangle = \langle\overline{w}''\overline{\theta}''\rangle. \tag{6}$$

The middle two terms in Equation 5 are zero via spatial averaging, since $\langle\overline{w}''\rangle = 0$ and $\langle\overline{\theta}''\rangle = 0$ by definition; therefore, these terms have no impact on spatially-averaged surface-atmosphere exchanges. Furthermore, $\langle\overline{w}\rangle$ is assumed to be very small (unless strong and large scale subsidence or uplift are present); in our LES it must be zero since there cannot be accumulation or depletion of mass below a given horizontal plane in a periodic domain with an incompressible flow. Thus, the first term on the right hand side of Equation 5 is also negligible, leading to one remaining term in Equation 6. This term, the dispersive flux, is of most interest: it represents the coherent spatial correlation of vertical velocity and potential temperature in regions with consistent secondary structures (such as consistent warm updrafts or cool downdrafts, or streamwise rolls).

Figure 5 shows the vertical profiles of the total, turbulent, and dispersive horizontally-averaged fluxes of each pattern for the heat flux, streamwise momentum flux, and cross-stream momentum flux, thus allowing us to decompose and analyze the total fluxes that were shown in Figures 4c-e. For example, it is clearer now that the cross-stream components of the total momentum flux (Figure 4e) are all distinct from one another due to these dispersive fluxes (dotted green lines). Over these heterogeneous surfaces, the dispersive cross-stream stress dominates over its turbulent in all patterns except Pattern5 (see panels c,f,i,l in Figure 5). The magnitudes of these dispersive fluxes are not equal when the heterogeneous surfaces are different from one another. Thus, it is not the ice fraction or average floe area, but the surface pattern itself, that leads to these differences in total cross-stream flux. The streamwise stresses, on the other hand, seem to be a balance of the dispersive and turbulent stresses that are not always of the same sign (middle column of Figure 5), and thus the dispersive components that directly result from the secondary motions imprinted by the surface pattern on the atmosphere are also critical here. These secondary circulations may be seen in the two-dimensional cross sections in the supporting information.

The total heat flux in all these simulations linearly decreases with height as dictated by the LES setup. As seen in Figure 4c, the variations are not as impressive as for the momentum fluxes, but can still result in a difference of up to 30%, especially near the surface. However, analyzing the left column of Figure 5 shows that the relative importance and profiles of the dispersive and turbulent heat fluxes exhibit more significant differences. The various surface patterns seem to lead to differences in the dispersive fluxes, but in all cases, these are balanced out by the turbulent fluxes. Nevertheless, in all the figures, there is strong variability near the surface in the dispersive and turbulent flux profiles. This is partially due to the shallow internal boundary layers that are created by the mean flow and secondary circulations over the floes, and it is these differing secondary circulations that arise due to the difference in surface patterns. However, one should also note that the first few points are strongly influenced by the transition from the wall model to the SGS model in representing resolved turbulence, a persistent

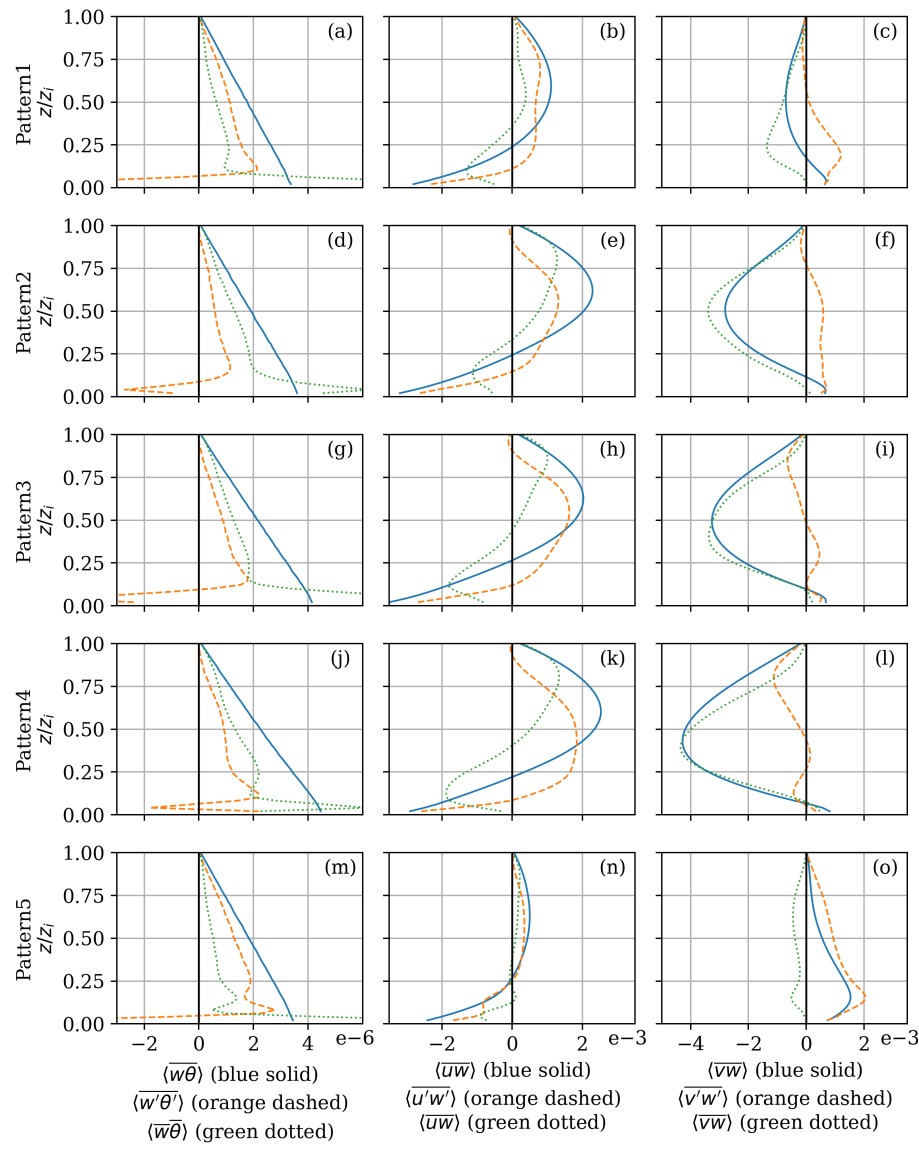

**Figure 5.** Left: Normalized (as in Figure 4) vertical profiles of total heat flux ($\overline{w\theta}$, blue solid), turbulent heat flux ($\overline{w'\theta'}$, orange dashed), and dispersive heat flux ($\overline{\widetilde{w}\widetilde{\theta}}$, green dotted) for each individual pattern. Center: Same as left, but for normalized total streamwise stress ($\overline{uw}$, solid), turbulent streamwise stress ($\overline{u'w'}$, dashed), and dispersive streamwise stress ($\overline{\widetilde{u}\widetilde{w}}$, dotted). Right column: Same as left, but for normalized total cross-stream stress ($\overline{vw}$, solid), turbulent cross-stream stress ($\overline{v'w'}$, dashed), and dispersive cross-stream stress ($\overline{\widetilde{v}\widetilde{w}}$, dotted).

**Table 2.** Dispersive-to-total atmospheric vertical flux ratios

|  | Pattern1 | Pattern2 | Pattern3 | Pattern4 | Pattern5 |
|---|---|---|---|---|---|
| $|\overline{w}\overline{\theta}|/|\overline{w\theta}|$ | 0.512 | 0.668 | 0.697 | 0.586 | 0.388 |
| $|\overline{u}\overline{w}|/|\overline{uw}|$ | 0.321 | 0.354 | 0.250 | 0.046 | 0.137 |
| $|\overline{v}\overline{w}|/|\overline{vw}|$ | 1.598 | 1.101 | 0.923 | 0.816 | 0.365 |

challenge in LES (Piomelli and Balaras, 2002; Brasseur and Wei, 2010); the quantitative details of the results in that region should be interpreted with care.

Lastly, analyzing the dispersive-to-total atmospheric vertical flux ratios (Table 2) lends insight into the differences between simulations, as well as comparison to other surface types. For example, in Pattern1 and Pattern2, $|\overline{v}\overline{w}|/|\overline{vw}| > 1$, meaning that the dispersive and turbulent fluxes in the cross-stream axis are in opposite directions, which can be seen in the green and orange profiles in the right column in Figure 5c,f. The ratios in Pattern4 and Pattern5 are also closer to unity than the other values, again showing the differences in secondary circulations due to the sea ice pattern. These values have been seen before, for example over urban or forest canopies (Moltchanov et al., 2015; Boudreault et al., 2017; Li and Bou-Zeid, 2019).

Overall, these LES results indisputably indicate that ice-water patterns hold key information on how the MIZ-ABL interacts with the underlying surface, and the rest of the paper is thus dedicated to characterizing these patterns

## 4 Results: Statistical Analysis

Now that we have established the need for additional surface characteristics beyond sea ice fraction, we aim to examine what indicators can be used for that purpose. For each of the nine resolutions considered, there were 44 observed sea ice images analyzed. Conducting this analysis over the $2\,\mathrm{m}$ resolution, four metrics (including sea ice fraction) seemed to remain from the VIF elimination process and not exhibit multicolinearity with one another: sea ice fraction ($f_i$), patch density ($PD$), splitting index ($SPLIT$), and perimeter-area fractal dimension ($PAFRAC$). These remaining metrics were then grouped into the different metric groups defined in Section 2.3: the sea ice fraction is an *area and edge* metric, $PD$ and $SPLIT$ are both *aggregation* metrics, while $PAFRAC$ is a *shape* metric. There were no metrics remaining that were in the *diversity*, *core area* or *contrast* metric group. We expected there to be no remaining metrics in the *contrast* metric group, since for a sea ice-water surface, an edge can only exhibit one "contrast" - however, this might change were this analysis to be conducted with continuously variable surface temperatures or roughness lengths. We also did not expect any from the *diversity* group, since many metrics such as evenness or Simpson's diversity index are functions of sea ice fraction. The absence of any representative of the *core area* was not predicted, but it is probably related to the fact that the *shape* and *area and edge* metrics are together able to represent the *core area* characteristics.

The first aggregation metric, PD (Riitters et al., 1995; Šímová and Gdulová, 2012), with a $VIF = 1.9$, is an area-normalized number of patches, described by the equation

$$PD = \frac{n}{A_t},$$ (7)

where $PD$ is the patch density, $A_t$ the total area of the surface, and $n$ the total number of distinct patches (of either sea ice or water). As the PD of a sea ice surface increases, one would expect to find more ice-water edge instances, and thus more regions of stable-to-unstable and unstable-to-stable stratification transitions. We also note that reducing $PD$ increases the average time the parcel spends over the stable (or unstable) surface, which affects how said parcel adjusts to the transition to this new stability regime. Patch density may also work in tandem with geostrophic wind direction, as discussed in Section 3, since a geostrophic wind flowing in one direction may have more ice-to-water edge transitions than another (see Pattern4 and Pattern5, for example).

The second metric, SPLIT, with a $VIF = 1.9$ was first described in Jaeger (2000). It represents the inverse of the probability that two randomly chosen points on the map will be in the same patch, with a corresponding equation of

$$SPLIT = \frac{A_t^2}{\sum_{i=1}^{n} a_i^2}$$ (8)

where $a_i$ is the area of patch $i$, and the index $i$ iterates over all patches. $SPLIT = n$ if there is only one patch or if all patches are of equal size. However, generally $SPLIT < n$, with lower values indicating a larger variance in patch sizes. Since we already use $PD$, the new information $SPLIT$ brings is precisely about the patch size variance.

The only shape metric, $PAFRAC$, with a $VIF = 2.1$ is obtained by regressing each patch's perimeter $P_i$ against its area $A_i$ on a log-log plot such that

$$A = kP^{2/PAFRAC},$$ (9)

where $k$ is a constant and $PAFRAC$ is the perimeter-area fractal dimension. It measures the tortuosity or jaggedness of the ice-water interface as with any fractal dimension (Mandelbrot, 1982).

Thus, the three metrics, in addition to sea ice fraction $f_i$, that would be useful in describing a sea-ice surface are $SPLIT$, $PD$, and $PAFRAC$. Table 3 details these values for each of the sea ice patterns simulated in Section 3. We observe that $SPLIT$ is invariant to the shape of floes as long as the area of each floe is equivalent (no variance), which is why $SPLIT$ is equivalent for all patterns except Pattern3. This is where $PAFRAC$ shows its utility, as it will give different values between Pattern1/Pattern2, Pattern3, and Pattern4/Pattern5.

For the real ice maps obtained from Arctic images, the ice fraction at a $2\,\mathrm{m}$ map resolution varies from 0.19 to 0.99, $PD$ from 5.4 to 42.5, $SPLIT$ from 1.23 to 4.07, and $PAFRAC$ from 1.338 to 1.726. In many cases however, numerical simulations also require a resampling of the high-resolution surfaces by increasing the grain (pixel) size. For example, sea ice maps from reconnaissance satellites may have a resolution up to $1\,\mathrm{m}$, but this is computationally impractical for numerical weather models. Large-eddy simulations of the ABL can have down to a $50\,\mathrm{m}$ resolution, while NWP models have $2$ to $10\,\mathrm{km}$ resolution. Therefore, even with high-resolution data, the aggregation and resampling of these surface patterns is inevitable

**Table 3.** Landscape metrics of the simulations conducted in Section 3. Note that for maps with a low number of patches (less than ten) and/or simple shapes, PAFRAC may exceed the theoretical range, as in Pattern3, but this has not happened in any of the real sea ice maps we examine next.

| Metric | Pattern1 | Pattern2 | Pattern3 | Pattern4 | Pattern5 |
|---|---|---|---|---|---|
| $f_i$ | 0.462 | 0.462 | 0.461 | 0.462 | 0.462 |
| Average Floe Area (m$^2$) | $11.56 \times 10^6$ | $11.56 \times 10^6$ | $11.56 \times 10^6$ | $11.56 \times 10^6$ | $11.56 \times 10^6$ |
| SPLIT | 2.91 | 2.91 | 2.71 | 2.91 | 2.91 |
| PD | $5 \times 10^{-8}$ | $5 \times 10^{-8}$ | $5 \times 10^{-8}$ | $5 \times 10^{-8}$ | $5 \times 10^{-8}$ |
| PAFRAC | 1.895 | 1.895 | 2.035 | 1.929 | 1.929 |

in modeling. Furthermore, when considering the operational use of these metrics, the regular updating of these values would likely draw upon multiple satellite products with differing resolutions; thus, metrics that are able to be extrapolated/interpolated between different grid cell sizes would allow for a consistent computation of metrics when standardized to a single weather model grid cell.

Therefore, it is useful to examine how these chosen metrics vary as an image is aggregated to a resolution applicable to numerical weather models (or other numerical models, such as LES); an appealing metric would be one that it is invariant to resolution changes. The sea ice fraction $f_i$ is a good example of such an invariant indicator, as shown in Figure 6a, where it is calculated for all images and then averaged over that resolution. A second-best case would be a metric that displays a clear scaling law with the resolution, such as PD depicted in Figure 6b. In this case, the PD for the "real" 2 m resolution surface can be extrapolated to higher resolutions based on a scaling power law

$$m = k\Delta^{D_q}, \tag{10}$$

where $m$ is the metric, $\Delta$ is the map resolution, and $k$ and $D_q$ are scaling coefficients.

Some metrics, such as $SPLIT$ and $PAFRAC$ seems to exhibit close-to-invariant behavior after a sort of 'jump' in the resolution. For example, starting from the 10 m resolution, $SPLIT$ stays fairly constant as the resolution decreases. There is also variation in $PAFRAC$ as the resolution decreases from 10 m. This is consistent with results from previous studies, as some landscape metrics exhibit large errors when these surfaces are aggregated to lower resolutions (Moody and Woodcock, 1994, 1995). Given the close-to-invariant scaling of 3 of the metrics and the predictable power law scaling of the forth, we can proceed with this set of 4 metrics since it is usable (or translatable) across scales.

## 5 Principal Direction of the Sea Ice Patterns

Thus far, we have identified four surface pattern indicators that characterize the MIZ surface's sea ice versus water concentration ($f_i$), the density and thus the total number of patches ($PD$), the variance in the sizes of these patches ($SPLIT$), and the tortuosity of their edges ($PAFRAC$). However re-examining Pattern4 and Pattern5 in Figure 4 raises the question as to why

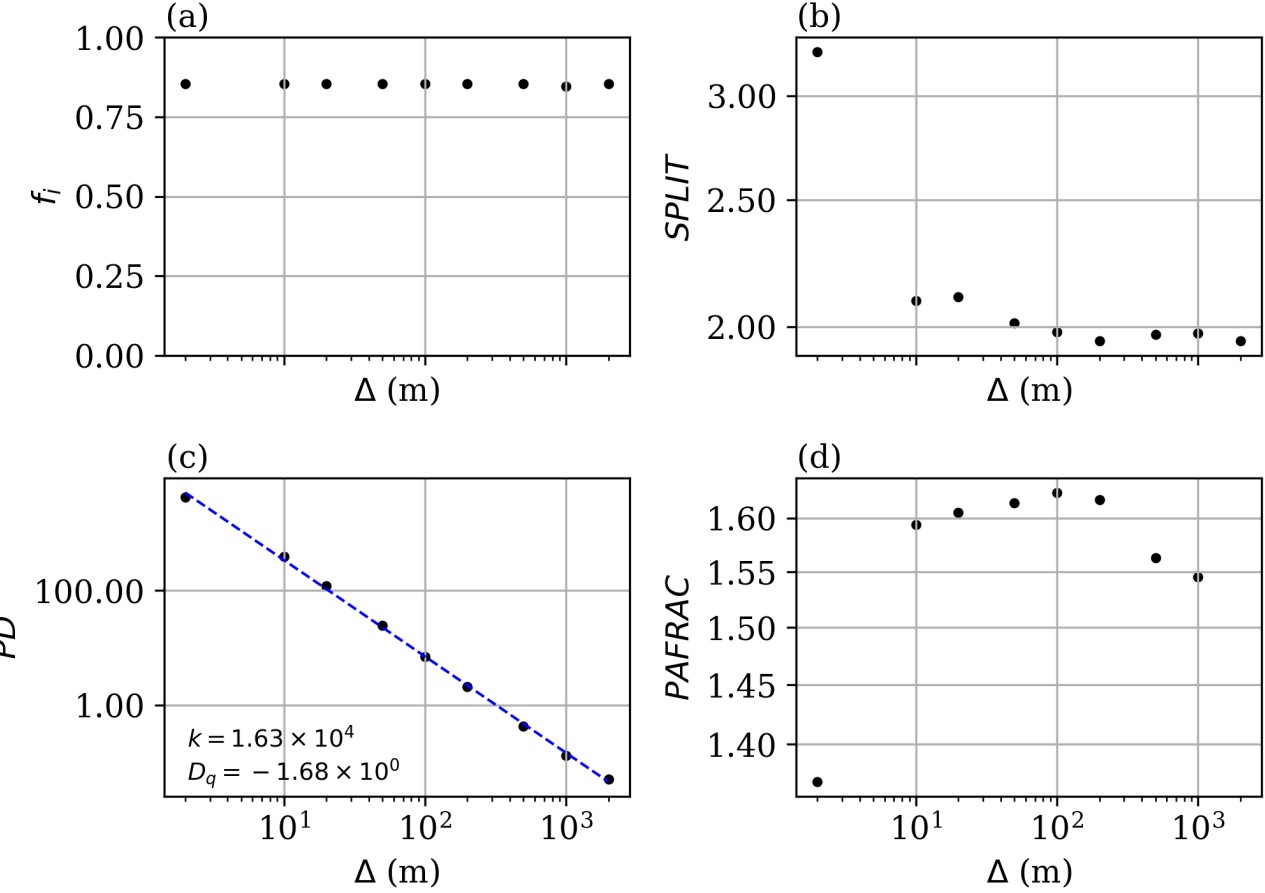

**Figure 6.** Landscape metric plotted against resolution for (a) sea ice fraction, (b) splitting index, (c) patch density, and (d) fractal dimension. A power law of the form $m = k\Delta^{D_q}$ is fitted to the patch density plot (dashed blue line), but no such power law is applicable to the other three metrics.

these maps have the largest differences in their respective MIZ-ABLs. Their geometric patterns are the same (thus the four metrics are identical for the two configurations), yet the difference in the geostrophic wind direction results in large differences in the surface-atmosphere interactions. This reveals that another important attribute is how the surface patterns are oriented relative to the wind. If the surface is isotropic, the wind angle should be irrelevant, but most water-sea ice patterns in the MIZ display a significant degree of anisotropy (Feltham, 2008). Therefore, quantifying the impact of surface orientation and including it in the metric set obtained from the VIF analysis in Section 4 may provide additional information for modelers to parameterize MIZ-ABL dynamics in global climate models.

We observe that in Pattern4 and Pattern5, the difference in the geostrophic direction is related to the directionality of sea ice organization. In Pattern4, the wind is consistently blowing over an infinitely repeating pattern of sea ice and water at regular intervals. In Pattern5, the wind blows over much longer strips of ice and water, even though there are some sea ice-water transitions present. Any other oblique flow is thus "in between" these two 'parallel' and 'perpendicular' regimes. We characterize the differences between these regimes by the variance of the surface that the wind is exposed to. In other words, Pattern4 exhibits a high variance since over one domain length, the surface wind flows over a maximum of eight ice-water transitions, while Pattern5 exhibits a low variance since the wind flows over a maximum of two sea ice-water transitions. This then raises the question of how to obtain some principal direction for a more complex surface.

We therefore attempted to characterize this anisotropy by computing the direction of the eigenvector (the eigendirection) of the surface with the least amount of variance, thus giving the fewest ice-water transitions possible. This was done by implementing the *scikit-learn* Python package, via a principal component analysis (PCA) using the `sklearn.decomposition.PCA` class (Pedregosa et al., 2011). This method performs the eigendecomposition of the covariance matrix of our sea ice map, yielding two orthogonal eigenvectors. The principal eigendirection points in the direction of minimal variance, denoted by the longer of the two arrows in Figure 7. That is, the principle or most coherent mode that best explains the pattern is along the direction of least variability. The secondary eigendirection is, by definition, orthogonal to the principal eigenvector. Some of these eigendirections are intuitive, as one can 'imagine' trying to pick a geostrophic wind direction that passes over a minimal number of ice-water edges. The maps in Figures 7a and 7g are two such examples. However, the map in Figure 7f, for example, is a bit less intuitive - visual inspection may suggest that the principle eigendirection should aligned from the lower right to the upper left, but the results indicate a less obvious orientation.

It is hypothesized that for a fixed $f_i$, the geostrophic wind flowing in a principal direction with minimal variance will behave more so like Pattern5, and the perpendicular angle to that principal direction (the secondary direction) will behave more so like Pattern4 (similar to the parallel and perpendicular cases in the simulations conducted in Fogarty and Bou-Zeid (2023)), but further LES simulations are needed to elucidate the exact impact of this relative orientation and the other parameters we identified on the MIZ-ABL.

However, some of these maps exhibit a higher degree of anisotropy than others, such as Figure 7c. To measure the degree of anisotropy in these maps, one may also look into percentage of variance (POV), defined as

$$\text{POV}(\lambda_i) = \frac{\lambda_i}{\sum_{i=0}^{n} \lambda_i}, \tag{11}$$

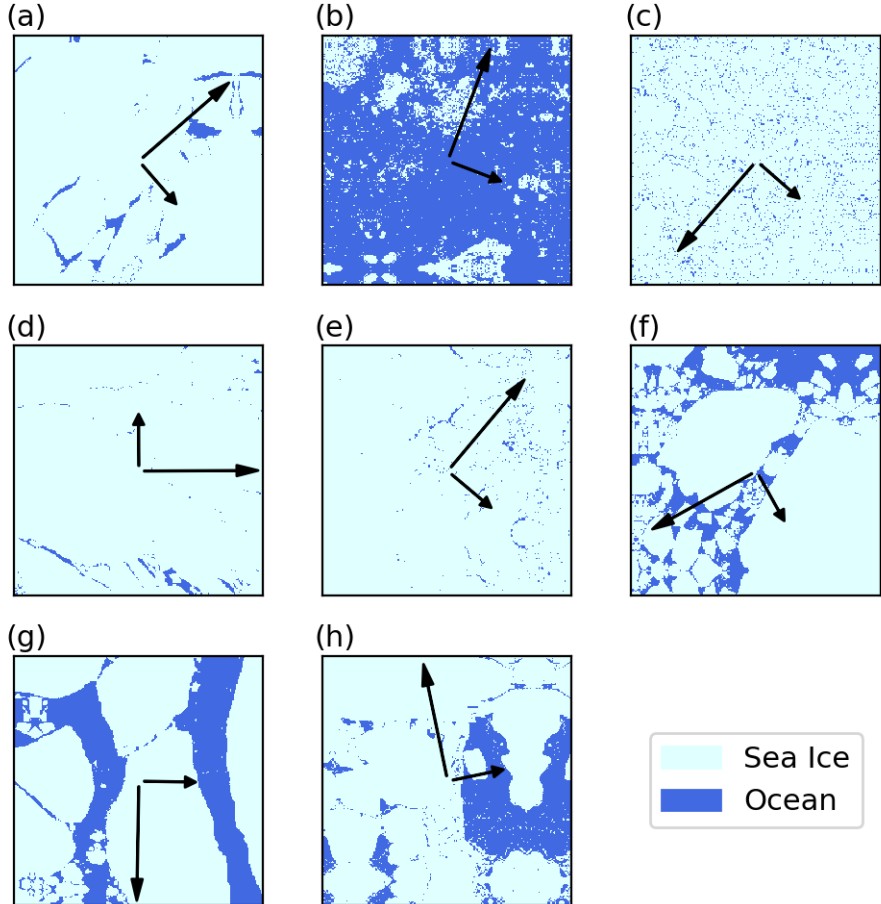

**Figure 7.** Eight maps from the sea ice dataset overlayed with their principal eigendirection (long arrow) and secondary eigendirection (short arrow), computed via principal-component analysis.

for an eigenvector $\lambda_i$ in an $n$-dimensional matrix. The POV of an eigenvector for a two-dimensional surface (n=2) thus describes the amount of variance that can be explained (or reconstructed) by that eigenvector alone, such that $\text{POV}(\lambda_0)+\text{POV}(\lambda_1) = 1$. In theory, a sea ice map with a high $\text{POV}(\lambda_0)$, and thus a low $\text{POV}(\lambda_1)$, would be anisotropic, since the secondary eigendirection would contain much more of the variance than the principal eigendirection, and the surface thus has a preferential direction of variability (one would expect Figure 7g to have a high $\text{POV}(\lambda_0)$). Conversely, a map with a low $\text{POV}(\lambda_0)$ would be a fairly isotropic map.

By definition, $\text{POV}(\lambda_0) \geq 0.5$ (with $\text{POV}(\lambda_0) = \text{POV}(\lambda_1) = 0.5$ resulting for a truly isotropic surface), since $\text{POV}(\lambda_0)$ is the POV for the principal eigendirection. However, most of the ratios in these examples lie in the range of $0.50 < \text{POV}(\lambda_0) < 0.60$,

**Table 4.** Eigenvector ($v_i$) angle and eigenvalue ($\lambda_i$) for principal ($i = 0$) and secondary ($i = 1$) eigenvectors, as well as percentage of variance (POV) of $\lambda_0$, corresponding to each map in Figure 7. Note that these angles are not traditional meteorological wind angles, but are instead in Cartesian coordinates; $0°$ is a left-to-right westerly wind, and $90°$ is a southerly wind.

| Map | $\angle v_0$ | $\lambda_0$ | $\angle v_1$ | $\lambda_1$ | POV($\lambda_0$) |
|-----|------|------|------|------|------|
| (a) | 41° | 3396 | 311° | 3286 | 0.508 |
| (b) | 69° | 4784 | 339° | 2688 | 0.640 |
| (c) | 229° | 3339 | 319° | 3330 | 0.501 |
| (d) | 0° | 3337 | 90° | 3285 | 0.504 |
| (e) | 50° | 3347 | 320° | 3332 | 0.501 |
| (f) | 209° | 3624 | 299° | 2688 | 0.574 |
| (g) | 269° | 3370 | 359° | 3293 | 0.506 |
| (h) | 102° | 3697 | 12° | 3033 | 0.549 |

showing little variability among these maps. Thus, in this small subset of sea ice maps, solely looking at POV($\lambda_0$) would not give information on how much influence the principal eigendirection has.

## 6  Conclusions

Although the stability over an ice- or water-dominated surface depends on many factors such as the wind direction and air potential temperature, for the cases where the air potential temperature falls between the surfaces temperatures of ice and water, the ice fraction of a sea ice surface itself can be fair indicator of the behavior of the MIZ-ABL; but only when the ice fraction approaches $0.0$ (all ocean, leading to an unstable atmosphere) or $1.0$ (all ice, leading to a stable atmosphere). On the other hand, when the $f_i$ is between these limits (in other words, if the surface flow alternates between very stable or very unstable), ice fraction alone is not enough to predict the dynamics and thermodynamics of the MIZ-ABL. Large-eddy simulations conducted for five different sea ice surfaces, detailed in Figure 2, have shown that surfaces with the same ice fraction, number of floes, and mean floe size can result in very distinct atmospheric dynamics. Differences were examined in the horizontal wind speed (Figure 4a) and total surface stresses (Figures 4d,e and 5).

While Figure 5 shows moderate differences in the total heat flux that is here constrained by the simulation setup, more significant differences are seen in the dispersive and turbulent fluxes that make up this flux (see also Table 2). The total, turbulent, and dispersive fluxes of the streamwise and cross-stream momentum were even more sensitive to the surface patterns. These dispersive fluxes are shown to drive many of the differences, and are thus non-negligible in climate models (Margairaz et al., 2020; Fogarty and Bou-Zeid, 2023; Lu et al., 2023).

To understand what other information one can obtain from a two-dimensional binary lattice surface, we examined 44 spatial metrics traditionally used in the field of landscape ecology, since knowing the cover fraction (ice fraction), and the number and median area of the floes is not enough to fully describe the ice-water-atmosphere physics. These 44 additional spatial

metrics were used on literal image derived products of real-world satellite sea ice imagery to determine which metrics were important, and the variance inflation factor was used to detect and remove multicolinearity in this dataset. The remaining metric set included ice fraction, patch density (representing the number of sea ice floes and thus their mean size in a given total area), splitting index (representing the variance in the floe sizes) and the perimeter-area fractal dimension (representing edge tortuosity). We also propose the use of the surface eigendirection relative to the mean wind direction, to characterize the influence of surface anisotropy and its interaction with the wind direction.

The resulting five metric set, including eigendirection, is not only useful for describing a two-dimensional surface, but based on the VIF analysis, it is also a minimal set of indicators needed to describe such a surface since they contain distinct and important information. However, the development of practical parameterizations for sea ice and the MIZ-ABL will ultimately need to include additional considerations including the ease of obtaining these parameters for modeling applications, the computing time needed to calculate these surface metrics dynamically versus resolving the surface features when running an ESM, the availability of easier-to-compute surrogate metrics, among others.

The first step in answering this broad question is investigating to what degree these other metrics affect the MIZ-ABL, in comparison with the first-order effect on the MIZ-ABL of ice fraction. In other words, given an ice fraction, how will changing any of the metrics in the resulting set affect the overlying MIZ-ABL? While this was answered in this study for idealized surfaces, proving that under some conditions these parameters are relevant, which of these parameters will be critical over real ice maps, and how often and to what degree, requires additional simulations (and a follow-up to this study is underway, see Fogarty et al. (2024)). More turbulence-resolving numerical simulations of the real ice surface are thus needed.

Another crucial step to answering this question is figuring out how one would go about creating an accurate parameterization based on the available external grid cell variables; the resources to answer this question may be extensive as well, especially considering that in this age of machine learning, high-resolution synthetic satellite imagery is being generated more often (see Au-Boehm et al. (2024)). Again, even more large-eddy simulations, beyond what has been done here, over real sea ice maps are imperative to answer this question. Lastly, this open question also requires looking at how to incorporate the resulting metrics and eigendirections into these climate models, such as examining (i) how the geostrophic wind at certain principal directions interacts with the resulting metric set, (ii) other possible metrics of anisotropy, and (iii) how the sea ice model in an ESM can provide the data needed to capture the heterogeneity of the sea ice surface - among other questions that remain unanswered at this time.

*Code and data availability.* A dataset containing the simulation results for the five patterns, and the FRAGSTATS output for the sea ice maps, are publicly available at https://doi.org/10.34770/5x2y-5485. FRAGSTATS is publicly available for download/use at https://fragstats.org/ (McGarigal and Marks, 1995), and the prepossessed sea ice maps are available at Fetterer et al. (2008).

## Appendix A: Large-Eddy Simulation Details

In this study, the incompressible filtered Navier-Stokes equations (with the Boussinesq approximation for the mean state) and heat budget are solved for a horizontally periodic flow, where a variable with a tilde represents a quantity filtered via the numerical grid spacing $\Delta$:

$$\frac{\partial \widetilde{u}_i}{\partial x_i} = 0 \,, \tag{A1}$$

$$\frac{\partial \widetilde{u}_i}{\partial t} + \widetilde{u}_j \frac{\partial \widetilde{u}_i}{\partial x_j} = -\frac{1}{\rho_r}\frac{\partial p}{\partial x_j} + \widetilde{F}_i + f_c \epsilon_{ij3} \widetilde{u}_j - g\delta_{i3}\left(1 - \frac{\hat{\theta}}{\theta_r}\right) - \frac{\partial \tau_{ij}}{\partial x_j} \,, \tag{A2}$$

$$\frac{\partial \widetilde{\theta}}{\partial t} + \widetilde{u}_j \frac{\partial \widetilde{\theta}}{\partial x_j} = -\frac{\partial q_j}{\partial x_j} \,. \tag{A3}$$

The equations above invoke the Einstein summation rule, where $i$ is the free index and $j$ the repeated index; $u_i$ is the velocity vector; $x_i$ is the position vector; $p$ is a modified pressure (see Bou-Zeid et al. (2005) for details); $\theta$ is the potential temperature; $\theta_r$ and $\hat{\theta}$ are, respectively, the Boussinesq reference (planar mean in our calculations) and the perturbation from that reference for potential temperature; $\rho_r$ is the reference mean density corresponding to $\theta_r$; and $F_i$ is the main flow-driving force (a synoptic pressure gradient). The Coriolis force is represented by the third term on the right-hand side of Equation A2, where $f_c$ is the Coriolis parameter and $\epsilon_{ij3}$ represents the Levi-Civita symbol. Buoyancy is represented by the fourth term on the right-hand side of Equation A2, where $\delta_{ij}$ represents the Kronecker delta.

An overbar denotes averaging in time, used as a surrogate for ensemble Reynolds averaging, while spatial averaging over the heterogeneous domain (in both $x$ and $y$) is denoted by angled brackets. The sub-grid scale stress $\tau_{ij} = \widetilde{u_i u_j} - \widetilde{u}_i \widetilde{u}_j$ and buoyancy flux $q_j = \widetilde{u_j \theta} - \widetilde{u}_j \widetilde{\theta}$, which result from the filtering, are modelled using a Lagrangian scale-dependent dynamic model (Bou-Zeid et al., 2005) with a constant sub-grid scale Prandtl number of $\mathrm{Pr} = 0.4$. As noted before, the numerical grid is the inherent filter of the model, but any explicit filtering needed to compute the dynamic Smagorinsky constant $c_s$ is done at scales $2\Delta$ and $4\Delta$ ($2\Delta$ for the local wall model); for these computations, a sharp-spectral cutoff filter is used. This model was validated by Bou-Zeid et al. (2005) for boundary layer flows over both homogeneous and heterogeneous terrain by reproducing experimental velocity and stress profiles obtained by Bradley (1968) after a change in surface roughness. It was then further validated for urban flows (Tseng et al., 2006; Li et al., 2016), and both stable and unstable boundary layers (Kleissl et al., 2006; Kumar et al., 2006; Huang and Bou-Zeid, 2013). Therefore, the ability of this model to successfully capture the impacts of stability and spatial transitions in surface properties was not tested further in this paper.

The LES employs boundary conditions that are periodic in the horizontal, with zero vertical velocity at the top and bottom of the domain, as well as a stress-free top lid ($\partial_z u_i = 0$ where $i = 1, 2$), with zero heat flux. These mimic a very strong top inversion, and are adequate for our setup since the top of the domain is not stably stratified and there is thus no need for a sponge to avoid wave reflection. This allows for the surface characteristics to be isolated from $z_i$ and the inversion strength. The indices $i = 1, 2, 3$ represent the $x$, $y$, and $z$ directions, oriented along the streamwise, cross-stream, and vertical directions, respectively. At the bottom of the domain, the surface stress and heat flux are computed by a wall model based on a local law-of-the-wall formulation (Bou-Zeid et al., 2005), with Monin-Obukhov buoyancy correction. Numerically, a pseudo-spectral approach is

employed in the horizontal, and an explicit second-order centred difference scheme used in the vertical. Time advancement is done using the fully explicit second-order Adams-Bashforth scheme. Dealiasing of the convective terms is performed using the 3/2 rule (Orszag, 1971). Pressure is computed from a Poisson equation obtained by taking divergence of the momentum equation and applying the incompressibility assumption.

## Appendix B:  Temperature Initialization

The initial potential temperature was chosen such that the mean heat flux over the entire domain is zero; in other words, the heat flux going into the ice is equivalent in magnitude to the heat flux coming from the water, based on the area fraction (ice fraction, in this case) of the domain. Thus, the initial air potential temperature of the large-eddy simulation was chosen such that the ice-fraction weighted heat flux over the ice ($f_i H_i$) was equivalent to the water fraction weighted heat flux over the ocean ($f_w H_w$),

$$-f_i H_i = f_w H_w \,, \tag{B1}$$

where $f_i + f_w = 1$. Using Monin-Obukhov flux profiles relations to express the surface fluxes:

$$\theta_i - \theta_a = \frac{H_i}{\kappa u_{*,i} \rho c_p} \left[ \ln\left(\frac{z}{z_{0h,i}}\right) - \Psi_s\left(\frac{z}{L_i}\right) \right], \tag{B2}$$

$$\theta_w - \theta_a = \frac{H_w}{\kappa u_{*,w} \rho c_p} \left[ \ln\left(\frac{z}{z_{0h,w}}\right) - \Psi_u\left(\frac{z}{L_w}\right) \right], \tag{B3}$$

where $\theta_a$ is the bulk air potential temperature; $\theta_i$ is the temperature of the ice surface; $\theta_w$ is the temperature of the ocean surface; $\kappa \approx 0.4$ is the von Kármán constant; $u_{*,i}$ and $u_{*,w}$ are the friction velocities of the ice and water surface, respectively; $\rho$ is the density of air; $c_p$ is the specific heat of air; $z$ is a height near the surface (taken at $z = 50\,\mathrm{m}$); $z_{0h,i}$ and $z_{0h,w}$ are the scalar roughness lengths of the ice and water surface, respectively; $L_i$ and $L_w$ are the Obukhov lengths over the ice and water surface, respectively; and $\Psi_s$ and $\Psi_u$ represent the stable and unstable correction functions, respectively, as reported in Brutsaert (2005). These Obukhov lengths are defined as

$$L_i = \frac{-u_{*,i}^3 \rho c_p}{H_i \kappa (g/\theta_a)}, \tag{B4}$$

$$L_w = \frac{-u_{*,w}^3 \rho c_p}{H_w \kappa (g/\theta_a)}, \tag{B5}$$

where the value of $H_i$ and $H_w$ in these equations is taken as a first-order estimate obtained by rearranging Equations B2 and B3 without the stability functions (i.e., for a neutral atmosphere),

$$H_i = (\theta_i - \theta_a)\kappa u_{*,i} \left( \ln \frac{z}{z_{0h,i}} \right)^{-1}, \tag{B6}$$

$$H_w = (\theta_w - \theta_a)\kappa u_{*,w} \left( \ln \frac{z}{z_{0h,w}} \right)^{-1}, \tag{B7}$$

which allows one to write a function substituting Equations B2 and B3 into Equation B1 to obtain a function that can be solved for $\theta_a$ via numerical root-finding. Of course, this is only a first guess to initialize the LES, which will then dynamically create its air potential temperature field during the warm-up period. As noted in the main text, the actual domain averaged heat flux in the LES will not be zero.

## Appendix C: Landscape Quantification Metrics

Table C1 lists the landscape metrics used in the VIF analysis conducted in Section 4. For more information on each individual metric (other than ice fraction), consult the FRAGSTATS manual (McGarigal and Marks, 1995).

**Table C1.** All landscape metrics used in the VIF analysis conducted in Section 4

| Ice Fraction | Edge Density | Interspersion and Juxtaposition Index |
|---|---|---|
| Number of Patches | Landscape Shape Index | Patch Cohesion Index |
| Patch Density | Perimeter-Area Fractal Dimension | Landscape Division Index |
| Largest Patch Index | Contagion Index | Effective Mesh Size |
| Total Edge | Percentage of Like Adjacencies | Splitting Index |
| Modified Simpson Evenness Index | Aggregation Index | Shannon Diversity Index |
| Simpson Diversity Index | Modified Simpson Diversity Index | Shannon Eveness Index |
| Simpson Evenness Index | | |

*Author contributions.* Joseph Fogarty: Conceptualization, Methodology, Formal analysis, Investigation, Writing - Original Draft, Visualization. Elie Bou-Zeid: Methodology, Investigation, Formal analysis, Writing - Review and Editing, Supervision, Funding acquisition. Mitchell
Bushuk: Writing - Review and Editing, Validation. Linette Boisvert: Writing - Review and Editing, Validation. All authors reviewed the results and approved the final version of the manuscript.

*Competing interests.* No competing interests are present.

*Acknowledgements.* This research was supported by the US National Science Foundation under award number AGS 2128345 and the National Oceanic and Atmospheric Administration, U.S. Department of Commerce under by award NA18OAR4320123. The statements,
findings, conclusions, and recommendations are those of the authors and do not necessarily reflect the views of the National Oceanic and Atmospheric Administration, or the U.S. Department of Commerce. We would also like to acknowledge high-performance computing support from Cheyenne (Computational and Information Systems Laboratory, 2019) provided by NCAR's Computational and Information Systems Laboratory, sponsored by the National Science Foundation, under projects UPRI0007 and UPRI0021.

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
