# Peer review of "How Many Parameters are Needed to Represent Polar Sea Ice Surface Patterns and Heterogeneity?"

_EGUsphere, 2024_

## Referee Comment (RC1)

Review of: "**How Many Parameters are Needed to Represent Polar Sea Ice Surface Patterns and Heterogeneity?**" by Joseph Fogarty, Elie Bou-Zeid, Mitchell Bushuk, and Linette Boisvert

Ian Brooks

**Overview & general comments**

This paper presents the results of an initial, and idealized, modelling study of the impact of the 2-dimensional spatial structure of broken sea ice and water surfaces on surface fluxes of momentum and heat for a fixed ice fraction. This is an important, but to date, little studied topic.

During much of the year, except for the summer melt, the skin temperatures of open water and sea ice can be very different, resulting in sudden changes in surface stability as air flows the boundaries between ice and water surfaces. Notably the air temperature – as in this study – may be at a temperature between that of the ice and water, so that stability changes from unstable over water to stable over ice. Depending on the length scale of continuous stretches of ice or water, the near-surface air may never have chance to adjust completely and for turbulence to reach a steady state. This has important implications for surface flux parameterizations in models, since the basis of these is Monin-Obukhov similarity theory, which relies on an assumption of statistical stationarity of turbulence and horizontal homogeneity of the surface.

Most surface flux parameterizations over sea ice are functions of ice fraction only (e.g. Elvidge et al. 2016, 2021), with some (e.g. Lüpkes et al. 2012, in its most complete form) including information about ice properties such as thickness, ridging, etc., which affect its roughness. Lüpkes and Gryanik (2015) have also attempted to address, from a theoretical perspective, the issue of changing stability with flow between ice and water surfaces.

This study does not directly address the issue of changing stability, although it is an inherent feature of the model set up, but examines a consequence – the fact that surface exchange, vertical profiles of turbulent fluxes, and the evolution of the vertical structure of the boundary layer, become functions of the spatial distribution of ice and water, even when the total fractions remain constant.

The results demonstrate that this dependence can be strong, and depends also on the orientation of the mean wind with respect to any anisotropy of the spatial patterns of ice and water. A key result is the importance of the diffusive flux contributions to the total flux – a result of spatially coherent correlations between vertical air motions and other quantities after time averaging.

One caveat that I think worth your discussing. While the results make clear the potential importance of the spatial distribution of ice/water surfaces, and the consequent potential importance of diffusive fluxes resulting from organised circulations induced by that spatial distribution, I think the cases presented here are likely to be extremes. The LES configuration has been chosen to ensure a surface flow parallel to the x-axis/east-west and to one axis of symmetry of the simple ice/water surface patterns. This ensures a nice, simple, well defined – and with periodic boundaries – repeating pattern of surface forcing. There is also a strong difference in surface temperatures for ice/water and hence the stability over the two surfaces. In all cases, part of the flow will encounter only a water surface (at north/south edges of domain), never flowing over ice, and thus always convective surface forcing, while the remaining flow will encounter varying fractions of alternating ice and water surfaces. This ought to encourage some sort of organised circulation to develop, forcing spatially coherent features in the time-averaged fields, and hence promoting the diffusive fluxes. This is unlikely to occur in this way in the real world. It doesn't invalidate the results, but it may make them rather extreme cases, and this should be noted.

Recommendation: publish with minor revisions

Elvidge, A. D., I. A. Renfrew, I. M. Brooks, P. Srivastava, M. J. Yelland, J. Prytherch, 2021: Surface heat and moisture exchange in the marginal ice zone: Observations and a new parameterization scheme for weather and climate models, *J. Geophys. Res.* 126, e2021JD034827, doi:10.1029/2021JD034827

Elvidge, A. D., I. A. Renfrew, A. I. Weiss, I. M. Brooks, T. A. Lachlan-Cope, J. C. King, 2016: Observations of surface momentum exchange over the marginal-ice-zone and recommendations for its parameterization. *Atmos. Chem. Phys.* 16, 1545–1563, doi:10.5194/acp-16-1545-2016

Lüpkes, C., Gryanik, V. M., Hartmann, J., and Andreas, E. L.: A parametrization, based on sea ice morphology, of the neutral atmospheric drag coefficients for weather prediction and climate models, J. Geophys. Res., 117, D13112, doi:10.1029/2012JD017630, 2012.

Lüpkes, C. and Gryanik, V. M.: A stability-dependent parametrization of transfer coefficients for momentum and heat over polar sea ice to be used in climate models, J. Geophys. Res. Atmos., 120, 552–581, doi:10.1002/2014JD022418, 2015.

**Detailed comments**

Line 16: "…shows persistent biases in coarse-resolution…" – need some additional detail, biases in…what? Ice fraction, thickness,…?

Table 1, and description of model set up (line 105-117) – most of this is fine, but I think a brief description of the initial air temperature is required. The way this is actually defined is technical, and sticking it in an appendix entirely appropriate, but I got lost when I reached the initial results. The formation of wind speed jets here requires a loss of frictional coupling to the surface to allow the low level wind to accelerate, and hence a stable stratification of the near-surface air…at which point I realized nothing had been said about the air temperature, and I had to go hunting through the text to see if I'd missed it. Eventually spotted the reference to appendix in Table 1, and could go and figure things out. But that really disrupts reading. Don't need a detailed description in the main text, but a brief statement along the lines of 'the initial surface air temperature is defined such that the area-averaged sensible heat flux is zero, and thus lies between that of the ice and water skin temperatures' (table 1 implies it is constant with height…is that temperature or (I assume) potential temperature?)

Figure 4: I assume the subscript 'g' on $M_g$ is for 'geostrophic' – this isn't noted in the text and should be stated for clarity.

Line 199: "All patterns except for Pattern5 developed a low-level jet" – it is pushing the definition of a low-level jet, but even Pattern5 does show a weak local maximum in the wind speed at $z/z_i$ ~0.3

Lines 200-202: discussion of low-level jets. The text states – "*The LLJs seem to increase in Pattern2 and Pattern3, likely due to large swaths of ice in the direction of the geostrophic wind (and therefore little interruption by the unstable ocean surface).*" – which is true, but I think it would be beneficial to expand on the discussion a little. A reader from the sea ice community rather than boundary-layer meteorology might not immediately appreciate that here a low-level jet forms only because of stable stratification below it, decoupling the air from surface friction, allowing it to accelerate and form a jet; and hence that over water, where the stratification is unstable, convective mixing would prevent a jet forming (or inhibit/erode a jet formed over the ice). I think a brief explanation of why the jet forms is needed, and perhaps more emphasis given to the very different stability over the ice/water surfaces.

Line 215 & figure 4: "…indicating significant differences in the wind and stress Ekman rotation with height" – might be useful to add a plot showing the wind direction profiles, to clearly show this.

Figure 5. Should make clear that altitude is normalized, = $z/z_i$ just to avoid any confusion.

Line 280: "one would expect to find more ice-water edge instances, and thus more regions of stable-to-unstable stratification transition" – while it is implicit in the set up, it is perhaps worth noting here that increasing patch density doesn't simply increase the number of stable-to-unstable stratification transitions, but also the unstable-to-stable, and importantly, reduces the time available over each consecutive surface type for the near-surface flow to adjust to the transition. At low PD, it is plausible that the flow approaches quasi-equilibrium, while for high PD that is never going to happen.

Line 290: it's not clear to me here exactly how PAFRAC is calculated…is the value just that of the gradient, *k,* in equation 8?

Line 300-307: This discussion presents the case for why you need to assess how the measures of ice/water distribution metrics behave when derived from images at different scales. I agree it is relevant and important to do this, but I think the rationale presented focuses on the wrong things.
The primary argument given here is that NWP models, and even most LES, have grid resolutions far lower than that of the high resolution imagery used here. True, but I think, irrelevant. The models don't require the raw imagery, only the metrics derived from it over an area of one grid cell, and ultimately the resulting transfer coefficients.
A more relevant issue, is that if this sort of information about ice/water spatial distribution is to be used operationally, then it will need to be updated on regular basis, and thus ideally draw upon all available imagery, each source of which may be at a different resolution. It is thus important to be able to achieve consistent results across different image sources.

Climate models are a different issue, since metrics must come from the sea ice model.

Figure 6: not clear why a power law scaling function is fit to the ice fraction when this is stated in the text to be scale invariant.

Figure 7. This shows ice/water maps derived from high resolution satellite imagery. Why are there clear regions of mirror symmetry, both horizontal and vertical (though curiously not always extending the full length of the image). I've marked the symmetries as red dashed lines on the copy of the figure below.

These may not affect the results (though ought to have a minor impact on the precise alignment of the eigenvectors), but they jump out as an oddity.

[Figure]

Lines 343, 358: At line 343 where the eigenvectors are introduced it is stated that "The principal eigendirection points in the direction of minimal variance", but then at 358 "the principal eigendirection explains much more of the variance" contradicting the first statement.

Lines358-362: "a sea ice map with a high POV($\lambda_0$) would be anisotropic", "Conversely, a map with a low POV($\lambda_0$) would be a fairly isotropic map", a 'low' value tends to imply $\rightarrow 0$, doesn't POV($\lambda_0$) $\rightarrow 0.5$ for isotropic conditions as implied by the statement "By definition, POV($\lambda 0$) > 0.5, since the POV($\lambda_0$) is the POV for the principal eigendirection.". What happens for a truly isotropic surface, where there is no preferential direction?

Table 4 caption: "Note that these angles are not traditional meteorological wind angles, but are instead in Cartesian coordinates, as 0∘ is a left-to-right westerly wind" – these are not wind angles (or rather, directions) at all. The important point is that the angles are stated in a Cartesian framework, increasing anticlockwise from the x-axis. They could be restated as compass headings…though it's not obvious here whether the ice maps are oriented north or with the field of view of the individual satellite orbits.

Line 457: "…such that the heat flux over the ice is equivalent to the heat flux coming from the water," – it's not clear from the text, but the implication of equation (B1) is that the initial temperature is chosen so that the mean heat flux is zero. 'equivalent to' is rather vague, be explicit.

**Minor typos & grammatical issues**

Line 6: "…such as those done in…" -> "…such as those used in…"

Line 28: "…it thus are…" -> "…it is thus…"

Line 42: "(as show for…" -> "(as shown for…"

Line 74: "…average pact compaction…" -> "…average patch compaction…" – I assume 'patch' but maybe something else was intended?

Line 130: "…here had already underwent…" -> "…here already underwent…"

Line 217: "…dispersive and turbulent counterparts" – 'components' or 'contributions' might be better words than 'counterparts' here.

---

## Author Comment (AC1)

**Responses to the review of: "How Many Parameters are Needed to Represent Polar Sea Ice Surface Patterns and Heterogeneity?" by Joseph Fogarty, Elie Bou-Zeid, Mitchell Bushuk, and Linette Boisvert**

Ian Brooks

**Overview & general comments**

This paper presents the results of an initial, and idealized, modelling study of the impact of the 2-dimensional spatial structure of broken sea ice and water surfaces on surface fluxes of momentum and heat for a fixed ice fraction. This is an important, but to date, little studied topic.

During much of the year, except for the summer melt, the skin temperatures of open water and sea ice can be very different, resulting in sudden changes in surface stability as air flows the boundaries between ice and water surfaces. Notably the air temperature – as in this study – may be at a temperature between that of the ice and water, so that stability changes from unstable over water to stable over ice. Depending on the length scale of continuous stretches of ice or water, the near-surface air may never have chance to adjust completely and for turbulence to reach a steady state. This has important implications for surface flux parameterizations in models, since the basis of these is Monin-Obukhov similarity theory, which relies on an assumption of statistical stationarity of turbulence and horizontal homogeneity of the surface.

Most surface flux parameterizations over sea ice are functions of ice fraction only (e.g. Elvidge et al. 2016, 2021), with some (e.g. Lüpkes et al. 2012, in its most complete form) including information about ice properties such as thickness, ridging, etc., which affect its roughness. Lüpkes and Gryanik (2015) have also attempted to address, from a theoretical perspective, the issue of changing stability with flow between ice and water surfaces.

This study does not directly address the issue of changing stability, although it is an inherent feature of the model set up, but examines a consequence – the fact that surface exchange, vertical profiles of turbulent fluxes, and the evolution of the vertical structure of the boundary layer, become functions of the spatial distribution of ice and water, even when the total fractions remain constant.

The results demonstrate that this dependence can be strong, and depends also on the orientation of the mean wind with respect to any anisotropy of the spatial patterns of ice and water. A key result is the importance of the diffusive flux contributions to the total flux – a result of spatially coherent correlations between vertical air motions and other quantities after time averaging.

One caveat that I think worth your discussing. While the results make clear the potential importance of the spatial distribution of ice/water surfaces, and the consequent potential importance of diffusive fluxes resulting from organised circulations induced by that spatial distribution, I think the cases presented here are likely to be extremes. The LES configuration has been chosen to ensure a surface flow parallel to the x-axis/east-west and to one axis of symmetry of the simple ice/water surface patterns. This ensures a nice, simple, well defined – and with periodic boundaries – repeating pattern of surface forcing. There is also a strong difference in surface temperatures for ice/water and hence the stability over the two surfaces. In all cases, part of the flow will encounter only a water surface (at north/south edges of domain), never flowing over ice, and thus always convective surface forcing, while the remaining flow will encounter varying

fractions of alternating ice and water surfaces. This ought to encourage some sort of organised circulation to develop, forcing spatially coherent features in the time-averaged fields, and hence promoting the diffusive fluxes. This is unlikely to occur in this way in the real world. It doesn't invalidate the results, but it may make them rather extreme cases, and this should be noted.

Thank you for your overall positive assessment and valuable comments. Regarding the last paragraph, the periodicity in our domain is chosen such that it represents the MIZ properly; in other words, the LES surface is an infinitely long domain with patches. Regarding the comment that these may be extreme cases, they indeed are by design. Our aim, which may not have been clear in version 1, is to show the maximum extent to which the spatial configuration of ice patches is important to the flow. We clarify in the revised version that in many real MIZ setups the effect would be smaller.

We also note that simulations in a previous paper (Fogarty & Bou-Zeid 2023) looked at circulations in two cases, one with consistently alternating strips of ice and water, and one with two parallel infinitely long patches of ice and water. In these cases, that convective surface forcing was not always present. Also in a forthcoming paper we will submit soon, we will be looking at real ice maps and examining how the configuration metrics developed here can help us estimate the effect of surface heterogenety.

**Recommendation: publish with minor revisions**

Elvidge, A. D., I. A. Renfrew, I. M. Brooks, P. Srivastava, M. J. Yelland, J. Prytherch, 2021: Surface heat and moisture exchange in the marginal ice zone: Observations and a new parameterization scheme for weather and climate models, J. Geophys. Res. 126, e2021JD034827, doi:10.1029/2021JD034827

Elvidge, A. D., I. A. Renfrew, A. I. Weiss, I. M. Brooks, T. A. Lachlan-Cope, J. C. King, 2016: Observations of surface momentum exchange over the marginal-ice-zone and recommendations for its parameterization. Atmos. Chem. Phys. 16, 1545–1563, doi:10.5194/acp-16-1545-2016

Lüpkes, C., Gryanik, V. M., Hartmann, J., and Andreas, E. L.: A parametrization, based on sea ice morphology, of the neutral atmospheric drag coefficients for weather prediction and climate models, J. Geophys. Res., 117, D13112, doi:10.1029/2012JD017630, 2012.

Lüpkes, C. and Gryanik, V. M.: A stability-dependent parametrization of transfer coefficients for momentum and heat over polar sea ice to be used in climate models, J. Geophys. Res. Atmos., 120, 552–581, doi:10.1002/2014JD022418, 2015.

Thank you for your responses and comments. The above citations have been added, and the comments below will be addressed.

**Detailed comments**

**Line 16: "…shows persistent biases in coarse-resolution…" – need some additional detail, biases in…what? Ice fraction, thickness,…?**

We mean biases towards predicted sea ice fraction and sea ice extent – this has been clarified in the text

**Table 1, and description of model set up (line 105-117) – most of this is fine, but I think a brief description of the initial air temperature is required. The way this is actually defined is technical, and sticking it in an appendix entirely appropriate, but I got lost when I reached the initial results. The formation of wind speed jets here requires a loss of frictional coupling to the surface to allow the low level wind to accelerate, and hence a stable stratification of the near-surface air...at which point I realized nothing had been said about the air temperature, and I had to go hunting through the text to see if I'd missed it. Eventually spotted the reference to appendix in Table 1, and could go and figure things out. But that really disrupts reading. Don't need a detailed description in the main text, but a brief statement along the lines of 'the initial surface air temperature is defined such that the area-averaged sensible heat flux is zero, and thus lies between that of the ice and water skin temperatures' (table 1 implies it is constant with height...is that temperature or (I assume) potential temperature?)**

Thank you for noting this disruption in reading, we have added: "The initial air temperature, a constant profile of potential temperature, $\theta$, is defined such that the area-averaged sensible heat flux is zero, and thus lies between that of the ice and water skin temperatures (see Appendix \ref{app:temp_init} for details),"

**Figure 4: I assume the subscript 'g' on Mg is for 'geostrophic' – this isn't noted in the text and should be stated for clarity.**

Added clarifications on the normalization on all the subplots in Figure 4

**Line 199: "All patterns except for Pattern5 developed a low-level jet" – it is pushing the definition of a low-level jet, but even Pattern5 does show a weak local maximum in the wind speed at z/zi ~0.3**

We edited to include Pattern 5 as well: "All patterns developed a low-level jet (LLJ), which can be seen in Figure \ref{fig:lesresults_4panel}a, though the LLJs in Pattern1 and Pattern5 are weak \citep{tetzlaff_aircraft_2015, michealis_convective_2021}." But we are a bit uncertain whether the reviewer is also referring to the fact that these are not the canonical LLJ people observe, and that may emanate from the Holton or Blackadar mechanisms (Du and Rotunno, 2014). Indeed this may be true; however as the reviewer postulates in the next comment, a Blackadar-like mechanism may be at play where parcels of air advecting from warm to cold patches experience decoupling from the surface and accelerate (this plays out in time in the original Blackadar version). Unfortunately, delving further into these physics and possibilities would not be feasible for this paper.

Du, Yu, and Richard Rotunno. "A Simple Analytical Model of the Nocturnal Low-Level Jet over the Great Plains of the United States." *Journal of the Atmospheric Sciences*, 2014. https://doi.org/10.1175/JAS-D-14-0060.1.

**Lines 200-202: discussion of low-level jets. The text states – "The LLJs seem to increase in Pattern2 and Pattern3, likely due to large swaths of ice in the direction of the geostrophic wind (and therefore little interruption by the unstable ocean surface)." – which is true, but I think it**

would be beneficial to expand on the discussion a little. A reader from the sea ice community rather than boundary-layer meteorology might not immediately appreciate that here a low-level jet forms only because of stable stratification below it, decoupling the air from surface friction, allowing it to accelerate and form a jet; and hence that over water, where the stratification is unstable, convective mixing would prevent a jet forming (or inhibit/erode a jet formed over the ice). I think a brief explanation of why the jet forms is needed, and perhaps more emphasis given to the very different stability over the ice/water surfaces.

Added a brief explanation on the formation of jets: "The LLJs seem to increase in Pattern2 and Pattern3, likely due to large swaths of ice in the direction of the geostrophic wind; the stable stratification in these ice regions decouple the air from the surface friction, allowing low-level acceleration of the wind (conversely, over an unstable ocean surface, the convection produced by the relatively warmer water inhibits this phenomenon). This mechanism is similar to the one advanced by Blackadar (1957) for creation of a low level jet via an inertial oscillation in time as the ABL transition to stable at sunset. However, in this case the oscillation is in space as columns of air advect from a hot to a cold surface and decouple from the surface"

Blackadar, Alfred K. "Boundary Layer Wind Maxima and Their Significance for the Growth of Nocturnal Inversions." *Bulletin of the American Meteorological Society* 38, no. 5 (January 1957): 283–90. https://doi.org/10.1175/1520-0477-38.5.283.

Line 215 & figure 4: "...indicating significant differences in the wind and stress Ekman rotation with height" – might be useful to add a plot showing the wind direction profiles, to clearly show this.

We've added to Figure 4 the wind direction profiles, right below the wind magnitude, to better understand the differences in wind turning with height.

Figure 5. Should make clear that altitude is normalized, = z/zi just to avoid any confusion.

Updated Figure 5 and its description: "Normalized vertical profiles of normalized total heat flux…"

Line 280: "one would expect to find more ice-water edge instances, and thus more regions of stable-to-unstable stratification transition" – while it is implicit in the set up, it is perhaps worth noting here that increasing patch density doesn't simply increase the number of stable-to-unstable stratification transitions, but also the unstable-to-stable, and importantly, reduces the time available over each consecutive surface type for the near-surface flow to adjust to the transition. At low PD, it is plausible that the flow approaches quasi-equilibrium, while for high PD that is never going to happen.

This is true. We've now noted this in the text: "As the PD of a sea ice surface increases, one would expect to find more ice-water edge instances, and thus more regions of stable-to-unstable and unstable-to-stable stratification transitions. We also note that $PD$ increases the average time the

parcel spends over the stable (or unstable) surface, which affects how said parcel adjusts to the transition to this new stability regime."

However, at low PD one might have some large patches that generate strong secondary circulations, while at very high PD it is also possible that the patch scale becomes so small that the atmosphere essentially sees a homogeneous average surface. So the interpretations may be more complex.

**Line 290: it's not clear to me here exactly how PAFRAC is calculated...is the value just that of the gradient, k, in equation 8?**

In the equation $A = kP^{2/D}$, it is $D$ that is PAFRAC. To keep it consistent with the other equations, we've replaced $D$ with PAFRAC in the equation and the text, and removed the variable '$D$' from the manuscript

**Line 300-307: This discussion presents the case for why you need to assess how the measures of ice/water distribution metrics behave when derived from images at different scales. I agree it is relevant and important to do this, but I think the rationale presented focuses on the wrong things.**

**The primary argument given here is that NWP models, and even most LES, have grid resolutions far lower than that of the high resolution imagery used here. True, but I think, irrelevant. The models don't require the raw imagery, only the metrics derived from it over an area of one grid cell, and ultimately the resulting transfer coefficients. A more relevant issue, is that if this sort of information about ice/water spatial distribution is to be used operationally, then it will need to be updated on regular basis, and thus ideally draw upon all available imagery, each source of which may be at a different resolution. It is thus important to be able to achieve consistent results across different image sources.**

**Climate models are a different issue, since metrics must come from the sea ice model.**

Thank you for your comment: overall, we think both ideas are true and important and we incorporated them since they bolster the ideas of the manuscript. We have thus added the following: "Furthermore, when considering the operational use of these metrics, the regular updating of these values would likely draw upon multiple satellite products with differing resolutions; thus metrics that are able to be extrapolated/interpolated between different grid cell sizes would allow for a consistent computation of metrics when standardized to a single weather model grid cell."

**Figure 6: not clear why a power law scaling function is fit to the ice fraction when this is stated in the text to be scale invariant.**

Good point, since ice fraction is scale invariant, we do not need to fit a power law (as seen by the value of $D_q$ being so low). We've changed the figure to only fit a power law to PD, and we've changed the caption to reflect this.

**Figure 7. This shows ice/water maps derived from high resolution satellite imagery. Why are there clear regions of mirror symmetry, both horizontal and vertical (though curiously not always extending the full length of the image). I've marked the symmetries as red dashed lines on the copy of the figure below. These may not affect the results (though ought to have a minor impact on the precise alignment of the eigenvectors), but they jump out as an oddity.**

Some of the original maps obtained from the dataset described in Section 2.2 were not full squares of data, i.e. there were ice cells, water cells, and 'no data' cells. To run this in LES in a square domain, we "reflected" the pattern over an axis of symmetry to preserve the pattern as best as possible. This only affected small border areas of the images. All map metrics, calculations, and analyses have been done on the "reflected" patterns. You are right that this does not affect the results of the VIF analysis, but may slightly affect the results of eigenvectors. We added a discussion on this in Section 2.2 – "Sea Ice Data", saying: "Some of the images did not fully cover this full extent, and thus in order to retain the real-world sea ice geometry, we "reflected" this onto the areas of no data. All metric calculations and analyses have been done on these modified surfaces."

**Lines 343, 358: At line 343 where the eigenvectors are introduced it is stated that "The principal eigendirection points in the direction of minimal variance", but then at 358 "the principal eigendirection explains much more of the variance" contradicting the first statement.**

Thank you for pointing this out, it should be the secondary eigendirection that contains more of the variance; we've edited this to say: "In theory, a sea ice map with a high $POV(\lambda_0)$, and thus a low $POV(\lambda_1)$, would be anisotropic, since the secondary eigendirection would contain much more of the variance than the principal eigendirection, and the surface thus has a preferential direction of variability"

**Lines358-362: "a sea ice map with a high POV(λ0) would be anisotropic", "Conversely, a map with a low POV(λ0) would be a fairly isotropic map", a 'low' value tends to imply →0, doesn't POV(λ0) →0.5 for isotropic conditions as implied by the statement "By definition, POV(λ0) > 0.5, since the POV(λ0) is the POV for the principal eigendirection.". What happens for a truly isotropic surface, where there is no preferential direction?**

For a truly isotropic surface (one where, let's say, the entire map is ice or the entire map is water), the program defaults to a left-to-right eigenvector, as it cannot pick a direction that stands out more than the others. And yes, the $POV(\lambda_0) = 0.5$ in this case – this has been clarified in the text.

Luckily, this doesn't present an issue operationally, since in a truly isotropic surface, there would no need to find such a "principal direction." We've added to the manuscript:

"By definition, $POV(\lambda_0) \geq 0.5$ (with $POV(\lambda_0) = POV(\lambda_1) = 0.5$ resulting for a truly isotropic surface), since $POV(\lambda_0)$ is the POV for the principal eigendirection."

**Table 4 caption: "Note that these angles are not traditional meteorological wind angles, but are instead in Cartesian coordinates, as 0∘ is a left-to-right westerly wind"** – these are not wind angles (or rather, directions) at all. The important point is that the angles are stated in a Cartesian framework, increasing anticlockwise from the x-axis. They could be restated as compass headings...though it's not obvious here whether the ice maps are oriented north or with the field of view of the individual satellite orbits.

Yes, the wind angles here do not follow meteorological convention, as 0 is left-to-right westerly wind, and 90 would be southerly. This has been clarified a bit further in the caption: "Note that these angles are not traditional meteorological wind angles, but are instead in Cartesian coordinates; 0° is a left-to-right westerly wind, and 90° is a southerly wind."

**Line 457: "...such that the heat flux over the ice is equivalent to the heat flux coming from the water,"** – it's not clear from the text, but the implication of equation (B1) is that the initial temperature is chosen so that the mean heat flux is zero. 'equivalent to' is rather vague, be explicit.

Yes, the interpretation is correct and this has been made clearer in the text: "The initial temperature was chosen such that the mean heat flux over the entire domain is zero; in other words, the heat flux going into the ice is equivalent in magnitude to the heat flux coming from the water, based on the area fraction (ice fraction, in this case) of the domain."

**Minor typos & grammatical issues**

All grammatical issues below have been resolved.

Line 6: "...such as those done in..." -> "...such as those used in..."

Line 28: "...it thus are..." -> "...it is thus..."

Line 42: "(as show for..." -> "(as shown for..."

Line 74: "...average pact compaction..." -> "...average patch compaction..." – I assume 'patch' but maybe something else was intended?

Line 130: "...here had already underwent..." -> "...here already underwent..."

Line 217: "...dispersive and turbulent counterparts" – 'components' or 'contributions' might be better words than 'counterparts' here.

---

## Author Comment (AC2)

**General Comments - Christof Lüpkes**

The marginal sea ice zone (MIZ) is characterized by strong surface inhomogeneity with respect to roughness and temperature. The typical scale of inhomogeneity is much smaller than the grid size of climate and weather prediction models, so that it is a challenge to parametrize turbulent fluxes close to reality. This paper attempts to study the impact of different ice floe patterns on domain averaged flux profiles over the MIZ by Large Eddy Simulation (LES).

The topic is challenging and important for polar climate modelling and weather prediction. In most parts the paper is well written, and the principal approach is adequate and can stimulate further scientific work. However, as explained below, there are some unclear points which should be considered before the paper is published. Qualitatively, the principle conclusions concerning the impact of sea ice patterns will probably not be affected by suggested modifications but their might be quantitative effects.

*We thank the reviewer for their overall positive assessment and valuable comments and suggested edits to our article. We address the specific comments and revisions below, and how they have been resolved in the manuscript.*

**Major revisions**

**My most important concern is the used grid size of 100 m. The problem of grid spacing in LES over sea ice with open water fraction is addressed in Gryschka et al. (2023), Lüpkes et al. (2008) and especially by Weinbrecht and Raasch (2001). The latter show that in LES 2 m grid spacing should be chosen when the width of open water leads is 200 m, Lüpkes et al. (2008) used in the LES 10 m for leads of 1 km width and a similar grid spacing (20 m) is chosen by Gryschka et al. (2023). Lüpkes et al. (2008) further show that for mesoscale simulations with coarser grid (200 m horizontal grid size) a lead specific nonlocal parametrization is necessary to obtain the correct plume inclination and vertical temperature gradients on the downstream side of open leads. The considered situation might be different due to the larger open water fraction but I recommend at least one model run with a strongly reduced grid size (e.g. 50 m) to test the sensitivity of the obtained flux profiles and thus main results on the resolution.**

*While we could reduce the size of the domain and increase the resolution, that will sacrifice some of the large scales and circulations that we are interested in capturing at these 10 km x 10 km extents.*

*We concur with the reviewer that the important parameter is the number of grid points used to resolve a patch, and this necessarily will deteriorate as patches get smaller and resolving each of them with ~ 100 grid points as in the papers cited by the reviewer become impossible computationally. However, a grid sensitivity test suggested by the reviewer is included below and shows us that there is not much change in the standard atmospheric variables (potential temperature and horizontal wind, see below). This sensitivity test is actually for a real ice map (since we had them readily available, it is for panel (b) in Figure 7 in the manuscript) and is run at 50*

m and 100 m resolution. As expected, the finer resolution results in more rapid warming of the atmosphere and thus higher heat fluxes at the surface as the simulations resolve the turbulence near the surface better. This then leads to a slight slowdown of the wind due to this stronger mixing.

[Figure]

[Figure]

We should note, however, that the LES simulations in this study are meant to demonstrate the need for surface analysis, given a constant ice fraction and average ice floe area. We thus do not focus on the quantitative aspect of the output. As such, we used a low resolution since it is sufficient to illustrate that configuration sensitivity. A manuscript is in progress where we continue this analysis by more closely examining the MIZ-ABL dynamics in response to surface heterogeneity – these simulations use a grid spacing of 50m.

**Furthermore, in chapter 2.2 it is said that resolutions of the sea ice maps are based on much higher resolutions (2m, 10m, 20 m etc.). I cannot follow here, the maps shown in Figure 2 do not reflect these resolutions. In case of a higher resolution of the surface than of the LES one would need subgrid scale flux parametrizations, which are not mentioned here. All this needs clarification.**

The different resolutions do not refer to Figure 2. The maps shown in Figure 2 correspond to Section 3 (Results: the MIZ-ABL over Idealized Configurations), where we created these configurations for the LES, at a resolution of 100 m. The description in Section 2.2, where we describe maps aggregated to resolutions of 2 m, 10 m, 20 m, etc. are real-world remotely-sensed maps that are not shown – these maps are used for the statistical analysis in Section 4 (Results: Statistical Analysis).

We have clarified this in the text, and furthermore, fixed the caption and content of Figure 2 to be clearer on what those five patterns are used for.

**The authors use a roughness length of 1 cm for open water. This value is much too high (orders of magnitude) for open water surrounded by sea ice. More reasonable values can be found, e.g. in Andreas et al. (2010), Lüpkes et al. (2012), Lüpkes et al. (2008), Elvidge et al. (2016), and**

**in Gryschka et al. (2023). The latter discussed the choice of 1 cm and consequences on results. Their findings concerning the effect of too large roughness lengths must at least be discussed but new simulations with realistic roughness length would be better.**

Thank you for your comment and the citations that we've added to the text. We recognize that our roughness length choices may not match what is in the current literature for the expected sea state in the MIZ. They corresponding to extremely wavy waters. Our aim, however, was to keep the roughness of ice and water equal to avoid injecting a roughness heterogeneity on top of the thermal heterogeneity. This is now clarified in the text. We may also add that the ice surface themselves can have roughness lengths that vary by order of magnitude (very smooth for flat ice and much larger for older wind battered ice).

We are in the process of preparing and submitting a new study of simulations over real ice maps. These simulations show that the roughness lengths do not make much of a difference between simulations. The thermal contrast between the ice and water have more of an effect on the results. See below for some preliminary results:

[Figure]

| Suite | $z_{0,i}$ (mm) | $z_{0,w}$ (mm) | $z_{0,i}/z_{0,w}$ | $z_{0,h}$ (mm) |
|-------|------|------|------|------|
| Red | 10 | 1 | 10 | 0.1 |
| Blue | 0.1 | 1 | 0.1 | 0.1 |

**It is not enough to show just the flux profiles and wind. To understand the consequences of assumptions, it is necessary to show also the domain averaged profiles of potential temperature as in Michaelis and Lüpkes (2022). Also vertical cross-sections of temperature and wind at some positions and horizontal cross-sections could be helpful to understand differences in the ABL structure between model runs as well as possible difficulties of the LES. It is also necessary to mention the height averaged wind speed in the ABL. At least in conditions with high ice fraction and some open leads Lüpkes et al. (2008) found a strong dependence of the ABL development on wind speed due to their importance on plume inclination over and downstream of leads.**

Thanks for your suggestion; we've added the domain-averaged profiles of temperature (along with wind direction, as per another reviewer) to the figure, and changed the resulting discussion. We have also added some horizontal and vertical cross sections as supplementary material.

**Lines 61-63: It seems that the authors are not aware of the papers Lüpkes et al. (2008), Michaelis et al. (2021) and Michaelis and Lüpkes (2022). In all papers it is explained that a parametrization for orthogonal flow over leads in sea ice is developed and applied based on LES. Thus, although it is not LES, it is qualitatively different to other mesoscale model applications. Especially interesting for the submitted paper of Fogarty et al. is the work of Michaelis and Lüpkes (2022). They do very similar studies applying their LES-based turbulence parametrization over an ensemble of leads (see e.g. their figures 6 and 7). The main difference is that the sea ice fraction is 93 %, so much higher than in the present study and that a simpler (2D) geometry of the open water fraction is used. The new findings of the present study should be discussed considering this work.**

Thank you for bringing our attention to these studies – we have added them to the manuscript (introduction section) and bolstered our discussion in the context of the findings of these papers. This helps our novelty in the sense that we are not only aiming for leads, but a mixture of leads and polynyas in the MIZ in the context of infinitely heterogeneous patches. Even if the LES accuracy could be improved with higher resolution and made more realistic with different $z_0$, these LES runs justify the need for the spatial analysis (which we get into in Section 4).

**A Coriolis parameter is used for 90°N. This needs justification because it is not really realistic. In winterly temperature conditions prescribed in the model, a sea ice fraction of 50 % would be a rare event at North Pole. A more realistic choice would be 80°N, the typical latitude of the MIZ in the Fram Strait. What is the effect of this choice?**

In the manuscript we have added the value of $f_c$ used in these simulations ($1.46 \times 10^{-4}$ s$^{-1}$), as well as the calculated Rossby number for all simulations.

$$\mathrm{Ro} = \frac{M_g}{f_c z_i} = \frac{(2 \text{ m s}^{-1})}{(1.46 \times 10^{-4} \text{ s}^{-1})(1000 \text{ m})} \approx 13.7$$

In effect, the value of $f_c$ is only important in relation to $M_g$ and $z_i$, as they jointly determine the Rossby number. One can then say that these simulations are valid for Rossby numbers of about 13.7 – since this is a dimensionless number, the inputs themselves can change without seeing a large change in the results (see Omidvar et al. (2020) and Allouche et al. (2022) – we also discuss this in your comment below).

Omidvar H, Bou-Zeid E, Li Q, Mellado J-P, Klein P. Plume or bubble? Mixed-convection flow regimes and city-scale circulations. Journal of Fluid Mechanics. 2020;897:A5. doi:10.1017/jfm.2020.360

Allouche, M., Bou-Zeid, E. & Iipponen, J. (2023) The influence of synoptic wind on land–sea breezes. Quarterly Journal of the Royal Meteorological Society, 149(757), 3198–3219. Available from: https://doi.org/10.1002/qj.4552

**The authors write always just 'air temperature'. But I think at all occurrences, they mean air potential temperature (e.g. in equations B2, B3). This means, however, that the model is initialized with a neutral stratification throughout the atmosphere. I am afraid that this might lead to unrealistic boundary layers. Note that the usually found is for such ice fractions a convective layer that is capped by a very strong inversion somewhere between about 300 and 700 m condition (if not affected by a thick stratus layer). Such inversions cause entrainment and influence the ABL development (see e.g. Tetzlaff et al., 2015). This needs at least discussion.**

Yes, when we write "air temperature," we do mean potential temperature since this is what our LES solves for – this has been clarified in the text. We fully agree that any other initialization will result in different results and we in fact have tested that. The present initialization is aimed to match the infinite (periodic) domain, assuming the air has been flowing over this type of pattern for a long time and is near equilibrium with that surface, with minimal effect from entrainment.

We simulate a strong capping at 1000 m, as there might not always be a strong convective layer. However, this capping depth will not affect these results, since in our LES one can rescale the domain down by half, for example, and obtain almost identical results (expect for the effect of increased Rossby). That is because our Reynolds number is effectively infinite given our MOST based wall model. One can also adjust $f_c$ to maintain a constant Rossby number. Because of this, we have expanded our discussion towards the end of the LES section to include dimensionless input parameters such as Rossby and Richardson numbers.

We also have a manuscript in progress with a principal focus on how the atmospheric dynamics and thermodynamics respond to surface heterogeneity.

**Minor revisions**

**Line 38: the term MIZ was introduced some decades before the paper of Dumont (2022), so that more references than just this paper should be given.**

True, so we've edited to say that we are just pointing to Dumont (2022) for a review on the current state of MIZ research

**Line 59: it should be even if some....**

Fixed

**Section 2.1: More information is needed here (see above): Which lateral boundary conditions are used in the LES? How strong is geostrophic wind? What about humidity? Are these dry runs without clouds?**

We've added the following details to Section 2.1:
- Geostrophic wind speed
- Horizontally periodic domain (also mentioned in Appendix A)
- Dry run with no clouds

**Figure 2: What is the unit of the axes? I suggest including two vectors illustrating the geostrophic wind and boundary layer wind.**

These patterns are 10 km x 10 km – updated in the figure, caption, and text.

**Line 136: I am not sure if I understood Figure 2 and its relation to the different resolutions correctly. This should be better explained. It would be helpful to use kilometers as a unit for the axes and to give some distances between floes (or the width of leads).**

Figure 2 does not have different resolutions associated with it; it is simply the idealized surfaces that were created for the LES portion of the manuscript. The multiple-resolution maps are used in the statistical analysis portion of the text. This has been clarified in the text, and Figure 2 has been updated to reflect the 10 km extent and geostrophic/surface wind vectors.

**Line 196: It is not the geostrophic wind alone. The near-surface wind is dominating the fluxes. However, the near-surface wind direction might differ from case to case for the same geostrophic wind.**

Thanks for the clarification, we've added: "and thus the near-surface wind" in the text.

**Line 200: One could cite Michaelis et al (2021) in this connection (occurrence of LLJ) as well as Tetzlaff et al (2015). This would support the results.**

Citations added, thanks for the suggestions

**Line 245: I would not write that differences are minimal. Note that smallest and highest surface fluxes differ by about 30 % from each other, which is a lot.**

We've changed the sentence to be: "the differences are not as impressive as the other variables, but can still result in a difference of up to 30\%, especially near the surface"

**Line 368: The stability over only ice or water depends on many factors, especially on the air temperature and wind direction. It can happen that there is an unstable stratification over sea ice (cold-air advection) and a stable stratification over the open ocean (warm air advection).**

We've added: "Although the stability over an ice- or water-dominated surface depends on many factors such as the wind direction and air temperature, for the cases where the air temperature falls between the surfaces temperatures of ice and water," at the beginning of this section.

**References**

Andreas, E. L., Horst, T. W., Grachev, A. A., Persson, P. O. G., Fairall, C. W., Guest, P. S., & Jordan, R. E. (2010). Parametrizing turbulent exchange over summer sea ice and the marginal ice zone. QJRMS, 136(649), 927-943.

Elvidge, A. D., Renfrew, I. A., Weiss, A. I., Brooks, I. M., Lachlan-Cope, T. A., & King, J. C. (2016). Observations of surface momentum exchange over the marginal ice zone and recommendations for its parametrisation. Atmospheric Chemistry and Physics, 16(3), 1545-1563.

Gryschka, M., Gryanik, V. M., Lüpkes, C., Mostafa, Z., Sühring, M., Witha, B., & Raasch, S. (2023). Turbulent heat exchange over polar leads revisited: A large eddy simulation study. J. Geophys. Res.: Atmospheres, 128(12), e2022JD038236.

Lüpkes, C., Gryanik, V. M., Witha, B., Gryschka, M., Raasch, S., & Gollnik, T. (2008). Modeling convection over arctic leads with LES and a non-eddy-resolving microscale model. J. Geophys. Res.: Oceans, 113(C9).

Lüpkes, C., Gryanik, V. M., Hartmann, J., & Andreas, E. L. (2012). A parametrization, based on sea ice morphology, of the neutral atmospheric drag coefficients for weather prediction and climate models. J. Geophys. Res.: Atmospheres, 117(D13).

Michaelis, J., Lüpkes, C., Schmitt, A. U., & Hartmann, J. (2021). Modelling and parametrization of the convective flow over leads in sea ice and comparison with airborne observations. QJRMS, 147(735), 914-943.

Michaelis, J., & Lüpkes, C. (2022). The impact of lead patterns on mean profiles of wind, temperature, and turbulent fluxes in the atmospheric boundary layer over sea ice. Atmosphere, 13(1), 148.

Tetzlaff, A., Lüpkes, C., & Hartmann, J. (2015). Aircraft-based observations of atmospheric boundary-layer modification over Arctic leads. Quarterly Journal of the Royal Meteorological Society, 141(692), 2839-2856.

Weinbrecht, S., & Raasch, S. (2001). High-resolution simulations of the turbulent flow in the vicinity of an Arctic lead. J. Geophys. Res.: Oceans, 106(C11), 27035-27046.